# Evaluating the Feasibility of Scaling the FIER Framework for Large-Scale Flood Inundation Prediction

Kel N. Markert[1,2], Hyongki Lee[3], Gustavious P. Williams[2], E. James Nelson[2], Daniel P. Ames[2], Robert E. Griffin[4], Franz J. Meyer[5]

[1]Google LLC, Mountain View, CA 94043, USA
[2]Department of Civil and Construction Engineering, Brigham Young University, Provo, UT 84602, USA
[3]Department of Civil and Environmental Engineering, University of Houston, Houston, TX 77204, USA
[4]Department of Atmospheric and Earth Science, University of Alabama in Huntsville, Huntsville, AL 35899, USA
[5]Geophysical Institute, University of Alaska Fairbanks, Fairbanks, AK, USA

*Correspondence to*: Kel N. Markert (kmarkert@google.com)

**Abstract.** Floods are a recurring global threat, causing lives lost, property damage, and agricultural impacts. Accurate and timely flood inundation forecasts are crucial for effective disaster preparedness and mitigation. However, traditional flood forecasting methods often face challenges in terms of computational demands and data requirements, particularly when applied to large geographic areas. This study presents a novel approach to scaling a data-driven flood forecasting framework, Forecasting Inundation Extents using REOF (Rotated Empirical Orthogonal Function) (FIER), to large geographic regions. FIER leverages historical satellite imagery and streamflow data to predict flood inundation extents offering a solution in regions typically considered data-scarce for traditional hydrodynamic modelling (i.e., lacking detailed bathymetry and friction coefficients information). We demonstrate the effectiveness of applying FIER over a large geographic extent using watershed boundaries to create individual FIER models and then mosaicking the results geographically to provide large flood inundation predictions. The Upper Mississippi Alluvial Plain in the United States was used as a test region. We evaluated multiple buffer sizes, ranging from 0 to 50km, for watersheds for generating the data-driven FIER models to reduce edge effects along watershed boundaries when mosaicking the individual FIER implementations. The FIER method using watersheds, coupled with different forecast lead times from the National Water Model operational streamflow forecasts, was used to accurately predict the extent of surface water for select flood and low flow use cases. Our results show that the scaled FIER approach using watersheds yields higher accuracies for different error metrics, including the Structural Similarity Index Measure (SSIM), RMSE, and MAE. We found that scaling FIER using a watershed approach yielded statistically significant better performance compared to the baseline area using the Kolmogorov-Smirnov test: this is particularly true when using buffer sizes for the watersheds of 0-10km and when applying a cumulative distribution function (CDF) matching post-processing correction to the FIER outputs. This approach offers a promising solution for large-scale flood forecasting, particularly in data-scarce regions where data required for traditional hydrodynamic modelling is lacking or ungauged basins. Future research will focus on refining the framework to incorporate additional hydrological variables and improve the accuracy of long-range flood inundation predictions.

# 1 Introduction

Natural disasters, with flooding the most prevalent, are estimated to cause over $300 billion in annual direct asset losses globally (Hallegatte et al. 2017). A recent study by the World Bank suggests that 1.47 billion people, or 19 percent of the world population, are directly exposed to substantial risks during 1-in-100-year flood events. Of these 1.47 billion people exposed to flood risk, 89 percent live in low- and middle-income countries (Rentschler et al., 2022). Climate change projections for 2030 indicate that the proportion of the population exposed to floods will increase (Tellman et al., 2021). Since 1980, 42 riverine and urban flooding events in the United States have cost a total of $197.2B (on average $4.4B per year) (Smith, 2020). Research has found that flood exposure and damages in the U.S. could also be exacerbated in the future due to anthropogenic climate change, population growth, and urban development (Tate et al., 2021; Wing et al., 2022). Accurate and timely forecasts that capture the spatiotemporal evolution of flood inundations with sufficient lead time for actions are crucial for mitigating the devastating impacts of floods on communities and infrastructure.

Hydrodynamic modeling is a widely used method for simulating the spatiotemporal behavior of flood inundation by creating inundation maps computed from modeled streamflow from hydrologic models (Teng et al., 2017). Modeling of streamflow using hydrologic models requires accurate forcing data, parameterization, and calibration which can lead to inherit errors in the predicted streamflow outputs (Renard et al., 2010). Furthermore, hydrodynamic models are highly sensitive to inputs, including the streamflow, the boundary and initial conditions, the digital elevation model (DEM) used, and friction coefficients, all of which are difficult to obtain and have associated variation and uncertainty. These uncertainties in hydrodynamic model calibration and data inputs significantly influence the uncertainty of flood inundation predictions (Bates et al., 2014; Teng et al., 2017) with the inundation extent estimates most sensitive to topography and friction coefficients (Yalcin, 2020). Hydrodynamic models carry a heavy computational burden, especially for a more accurate high-resolution large-scale forecasting framework, that could affect forecast lead-time and accuracy (Ben-Haim et al., 2019). While continental-scale hydrodynamic models such as LISFLOOD-FP (Sampson et al., 2012), CaMa-Flood (Yamazaki et al., 2011), or HyMAP (Getirana et al., 2012) are more computationally efficient and have been successfully implemented at large scale, they are typically run as "offline" models or are set up to run at a coarse resolution (1 - 25 km resolution) which limits the use for operational flood inundation purposes at the local level. Even these more efficient models still require detailed parameterization, which can introduce errors, making them impractical in some cases due to data requirements, uncertainty, and complexity.

An area of active research in flood forecasting leverages advancements in Earth observations and machine learning to enhance prediction accuracy and provide spatially explicit inundation information. Data-driven approaches are being explored to establish relationships between rainfall forecasts, satellite-observed inundation patterns, and other hydrometeorological variables, enabling more efficient and potentially accurate flood extent predictions. For example, a recent study published methods for forecasting inundation extent using Earth observation data, such as rainfall forecasts with machine learning approaches to estimate water fraction (Du et al., 2021). Another example that uses machine learning to

estimate flood inundation extent is the Google Flood Forecasting system (Nevo et al., 2022), which trains a per-pixel thresholding algorithm on historical satellite observations of flooding using simulated streamflow as inputs. These research efforts integrating historical satellite data, machine learning algorithms, and hydrologic variables highlight a promising avenue for developing robust flood forecasting systems.

70  A promising data-driven framework for predicting surface water extents using satellite and hydrologic data is the Forecasting Inundation Extents using Rotated Empirical Orthogonal Function (FIER) framework (Chang et al., 2020). This framework operates by extracting historical patterns to identify recurring spatial and temporal patterns of flooding using a statistical technique called Rotated Empirical Orthogonal Function (REOF) analysis (Kaiser, 1958). Then these flood patterns are then correlated with historical hydrological data (e.g., streamflow, water levels) to build regression models. Using simulated

75  (retrospective or forecast) hydrological data as input, FIER synthesizes corresponding flood maps. FIER allows user to simulate inundation maps without the need to develop and calibrate a complex hydrologic and hydrodynamic model from scratch, instead utilizing existing, often operational, streamflow forecasts. The main advantages of this framework are its computational efficiency, scalability, and ability to operate in data-scarce regions. While relying on existing modelled streamflow from operation systems can be advantageous, users also inherit the errors from operational model outputs which

80  can lead to some performance degradation in the outputs. To date FIER implementations are typically trained on and applied to specific regions, limiting their applicability to broader areas. Hydrological regimes, topography, and flood characteristics vary significantly across different geographical locations, requiring regionally tailored implementations for accurate predictions. This is due to FIER being a data driven method meaning that the method is dependent on the data inputs and the patterns it can extract from the data. FIER has been applied to the Mekong Delta (Chang et al., 2023) and small regions in

85  the US (Rostami, et al., 2024) but it is unknown how the method will perform when attempting to develop the model for very large areas (e.g., all of the Mississippi basin) when there may be varying patterns of floods. Moreover, there is a computational challenge as the nature of developing the flood patterns requires loading data in memory for processing so applying FIER over large areas can be a challenge. Expanding the spatial coverage of FIER is crucial for the applicability of the method for an operational product over large areas.

90  This paper explores the feasibility of applying FIER in a manner that creates a consistent inundation forecast for both flooding and low-flow cases for large areas making the methods applicable for operational use. We apply FIER for multiple watersheds across a large area and test combining them to create a seamless surface water predictions. The results of the method are compared against a baseline implementation of FIER for a single area to compare how the combination for large area simulation compares to the traditional methods. The technical implementation and statistical validation are described.

95  Furthermore, we provided additional analysis to highlight the method's capability to provide accurate flood forecasts. This approach has the potential to be highly beneficial for operational flood forecasting and can contribute to more effective decision-making in the events of floods. By enabling the application of FIER to large river basins with diverse hydrologic characteristics, this research paves the way for developing robust, computationally efficient inundation forecasting systems worldwide, particularly in data-scarce regions where traditional hydrodynamic models are often infeasible.

## 2 Details on the FIER framework

The FIER framework offers a novel approach to flood inundation forecasting, leveraging a data-driven method to produce spatial flood inundation estimates without the complexities and computational needs of traditional hydrodynamic models such detailed data, parameterization, and calibration for successful implementation. While FIER involves its own data processing and model fitting steps, the framework shared as an open-source Python package is designed to be accessible. At its core, FIER establishes a statistical relationship between historical flood patterns, derived from satellite imagery, and corresponding hydrological data, typically streamflow or water levels. This relationship is then used to predict historical and future flood extents based on modeled hydrological conditions.

The FIER process begins by applying Empirical Orthogonal Functions (EOF) (Lorenz, 1956) to a multi-temporal stack of satellite images that capture historical flood events. EOF, a variant of Principal Component Analysis (PCA), decomposes the spatiotemporal variability of the images into a set of orthogonal spatial patterns, and their corresponding temporal variations. The original images can be reconstructed by weighted combinations of these components. In many cases the signal from any individual component may not be significantly different from random noise, therefore a Monte Carlo significance test (Hannachi, 2004) is performed to identify the significant components. The extracted significant components that are retained represent truncated information. In some fields physical meaning can be assigned to components, but the extracted significant components may not contain isolated signals, meaning individual components are hard to interpret as physical processes (Dommenget and Latif, 2002), therefore a rotation is applied, in this case the varimax rotation, which changes the orientation of the factors without altering their fit to the data to obtain simple structures. This makes it easier to physically interpret the inundation patterns in the components. The process of rotating the EOF is known as Rotated Empirical Orthogonal Function (REOF) analysis. The resulting spatial patterns are termed Rotated Spatial Modes (RSMs) whereas their corresponding temporal variations are called Rotated Temporal Principal Components (RTPCs). Each RSM represents a distinct spatial pattern, while its associated RTPC describes how that pattern evolves over time. Next, a correlation analysis is performed that identifies the RTPCs that are significantly correlated with the hydrological data, representing flood-relevant modes. Regression models are then built to link these flood-relevant RTPCs to the corresponding hydrological variable. These regression models can consist of using generalized linear models (Chang et al., 2020) or more sophisticated machine learning/deep learning approaches (Chang et al., 2023). To forecast flood inundation, forecasted hydrological data are used as the input into the trained regression models to predict future RTPCs. These predicted RTPCs are then multiplied by their corresponding RSMs and summed to synthesize a forecasted flood signal (essentially a reverse PCA), which can be further processed to generate a map of predicted flood extent. Figure 1 displays a flowchart schematic which summarizes the FIER process, readers are directed to (Chang et al., 2020, 2023) for additional details on the FIER process.

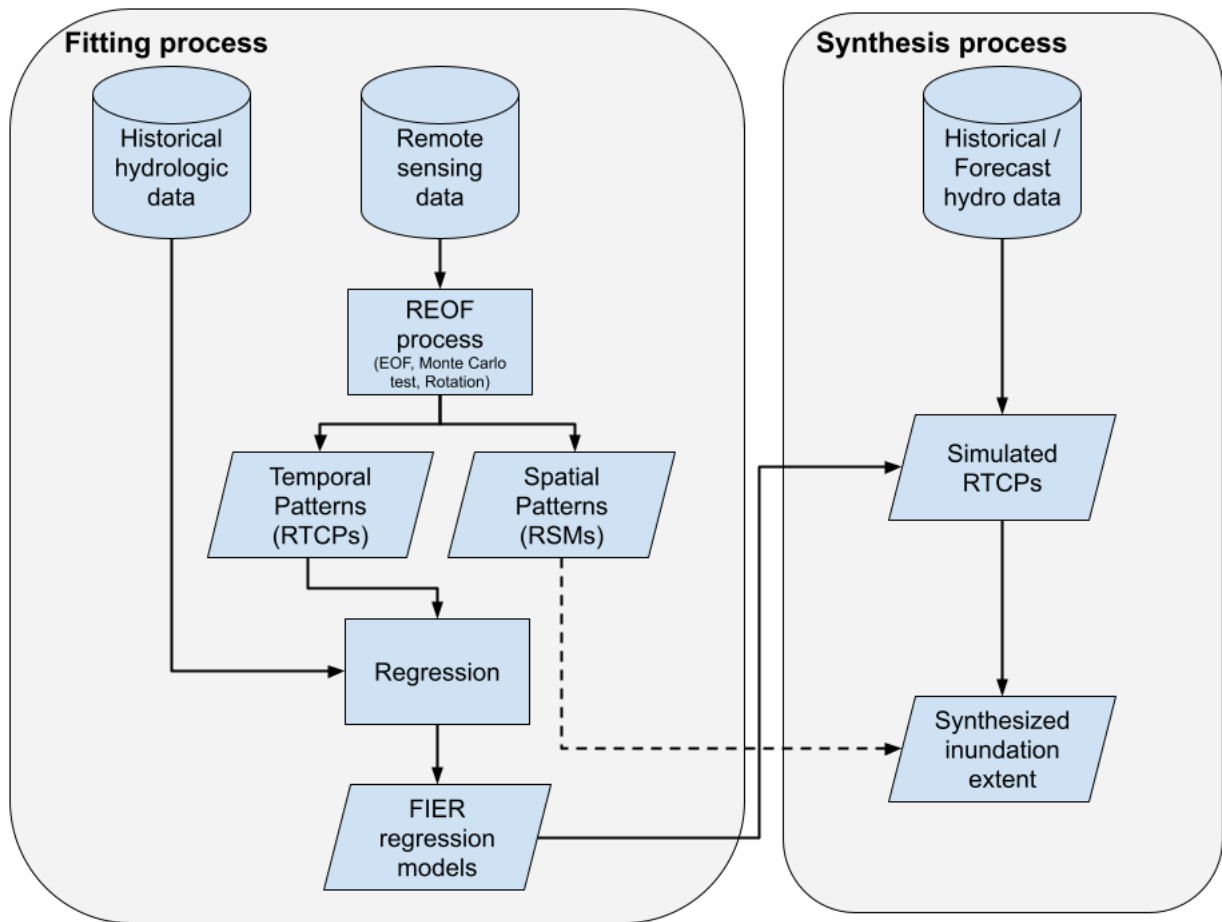

**Figure 1: Schematic of the FIER workflow. Adapted from (Rostami, et al., 2024).**

135    Scaling the FIER approach to larger geographic extents presents several challenges. One key limitation is the diminishing signal of floods as the area of analysis increases. FIER relies on identifying recurring spatial patterns of flooding from satellite imagery. As the area expands, these patterns become less distinct and more challenging to extract, particularly in regions with diverse hydrological regimes or where flooding is not widespread. This can lead to reduced accuracy and difficulty in establishing robust relationships between flood patterns and hydrological variables because the REOF process

140    may extract other signals occurring on the land surface.

Additionally, computational challenges arise when processing large volumes of satellite data and performing REOF analysis over extensive areas. The FIER training implementation requires significant computational resources and processing time when applied to large areas at regional to continental scales. However, applying predictions using the framework are

relatively fast. For example, applying the REOF process part of the FIER training over large areas requires reading in an entire time series of satellite imagery into memory and applying the PCA and REOF process over massive arrays. This challenge limits its applicability to select organizations with such computational resources and hinders its operational feasibility due to computational needs and runtime. These limitations necessitate exploring alternative approaches, such as the proposed watershed-based scaling method, to overcome the diminishing flood signal and computational bottlenecks associated with scaling FIER to larger geographic extents.

## 3 Materials & Methods

### 3.1 Study Area

This study focuses on a flood plain as part of the lower Mississippi basin, more specifically the region extending from approximately St. Louis, Missouri past Memphis, Tennessee. This region was selected because it is characterized by extensive floodplains along the Mississippi River and complex network of tributaries which experience flooding and droughts. The primary inflows to this segment of the Mississippi are from its own upstream reaches and major tributaries like the Missouri River (joining upstream of the study area near St. Louis) and the Ohio River (joining near the downstream end of our baseline area at Cairo, Illinois). The Cumberland and Tennessee Rivers are major tributaries to the Ohio, contributing significant volume to the system.. Furthermore, there are reservoirs, namely the Kentucky Lake, Lake Barkley, and Pickwick Lake on the eastern side, Rend Lake in the north, and Sardis Lake to the south within the region which are used for hydropower generation and to regulate flow into the Mississippi River to reduce flooding. Additional structures along the study area include dikes and levees constructed by The US Army Corps of Engineers for flood control (Watson et al., 2013). The Mississippi River near the inflow of the study area measured at the USGS gauge in St. Louis has an average streamflow of 6,197 $m^3$/s with a peak streamflow ranging from 29,166. – 101,940 $m^3$/s for years 2012-2020. Flood events in the lower Mississippi area are primarily triggered by rainfall and snowmelt. More recently, the region is experiencing a shift in climate leading to increases in streamflow (Yin et al., 2023). In 2011 and 2019 there were flooding events related to heavy precipitation and late spring snowmelt (Gledhill et al., 2020) where the recent flood event in 2019 was regarded as one of the longest lasting events in the past century (Pal et al., 2020). Conversely to the reported long-term climatic changes, the Mississippi River also experiences significant droughts with the most recent record-low water levels being recoded in 2022 and 2023, particularly in the Memphis area. Droughts and the low water levels are caused by a lack of precipitation in the Mississippi River basin and extreme temperatures contributing to excessive evapotranspiration (Muñoz et al., 2023). All these factors lead to a complex hydrology for the region making it an ideal candidate for testing the scalability of the FIER method.

Figure 2 shows the study area along with historical average (2012-2020) surface water fraction derived from the Visible Infrared Imaging Radiometer Suite (VIIRS) sensors. The study uses two subsets of the broader region for testing the methods: 1) a baseline region and 2) the watersheds surrounding the baseline region (see details in section 3.3 Experimental

Design for further explanation on the two subregions). The baseline region is marked in red whereas the watersheds are shown with the black outlines.

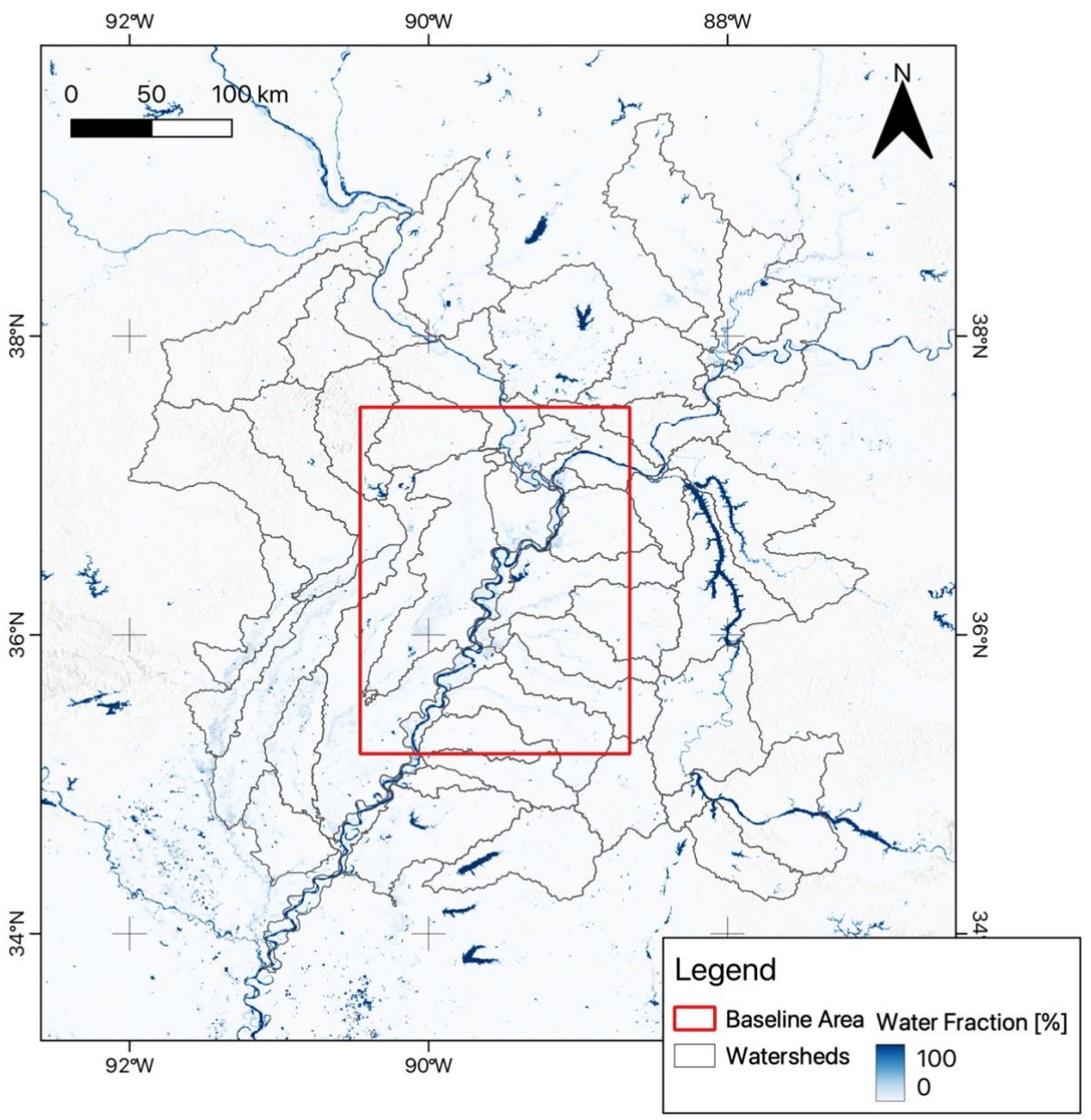

**Figure 2: Study area map showing the average surface water fraction derived from the VIIRS sensor from 2012-2020 along with**
180 **two subregions, the baseline region for FIER in red and the surrounding watersheds in black.**

## 3.2 Data

The Visible Infrared Imaging Radiometer Suite (VIIRS) is an optical sensor onboard the Suomi-NPP, NOAA-20, and NOAA-21 satellites. The VIIRS sensors provide images with a spatial resolution of 375 meters across five spectral bands ranging from visible to thermal infrared channels and has a swath width of 3,000 km with a consistent across-scan spatial resolution. These sensor characteristics ensure more comprehensive global daily coverage and make it more favorable than the MODIS data for flood mapping (Li et al., 2020, 2022). The National Oceanic and Atmospheric Administration (NOAA) uses the VIIRS imagery from the three satellites to produce an operational daily surface water fraction estimate for the entire globe (Li et al., 2018). The VIIRS water fraction product produced by NOAA was used to extract the spatiotemporal patterns of surface water and flood changes using the FIER framework. These data were accessed from the AWS Registry of Open Data, specifically the NOAA Joint Polar Satellite System (JPSS) cloud storage bucket (https://registry.opendata.aws/noaa-jpss/). The data were ingested into Google Earth Engine (Gorelick et al., 2017) as an ImageCollection. Earth Engine was used to preprocess the VIIRS data and extract data cubes in the format required for processing with FIER. We used the full record of VIIRS water fraction maps for the study, 2012-01-20 to 2023-12-31, however the VIIRS water fraction data has a missing period from 2021-01-01 to 2023-08-10 that was excluded from the study. While SAR imagery offers cloud penetration advantages and has been used with FIER (Chang et al., 2023), VIIRS was selected for this study due to its extensive and consistent daily historical archive dating back to 2012, which is crucial for extracting robust spatio-temporal patterns via REOF over a long period and diverse hydrological conditions (Rostami et al., 2024).

We used the National Water Model (NWM) streamflow data as the hydrologic variable to predict the flood-relevant temporal patterns with FIER. The NWM (Cosgrove et al., 2024) is a hydrologic modeling framework developed by NOAA at the National Water Center in Tuscaloosa, Alabama that simulates streamflow data for over 2.7 million river reaches across the United States along with other hydrologic information such as snow water equivalent and soil moisture. The NWM forecasts include two datasets, the retrospective dataset which is a historical simulation from 1979 to 2023 and the operational dataset which is run every day to produce operational forecasts since late 2018. The operational dataset includes a short-range 18-h deterministic forecast that provides flow estimates on an hourly time step, a medium-range forecast with 10 days (member 1) and 8.5 days (members 2–6) with a 3-h time step and a long-range 30-day four-member ensemble forecast on a 6-h time step. The short-range forecast is initialized every hour and the medium- and long-range forecasts are initialized every 6 h. In addition to the forecast runs there is also an analysis and assimilation run which is a nowcast of current streamflow conditions that includes data assimilation from streamflow gauges and a run with no data assimilation. We used both NWM datasets in this study; the retrospective data version 3.0 were used to fit the FIER temporal components to the historical simulated streamflow whereas the operational products were used to predict flood extent for select cases. The retrospective data was accessed through the AWS Registry of Open Data from the NOAA National Water Model CONUS Retrospective Dataset (https://registry.opendata.aws/nwm-archive/). The operational data was accessed via the Google Cloud Public Dataset on BigQuery (Markert et al., 2024b).

## 3.3 Experimental Design

The experimental design for this study aims to evaluate the effectiveness and accuracy of scaling the FIER method across larger spatial extents. We tested two approaches: applying FIER to a singular baseline area encompassing a portion of the study area, which is used as the control, and applying FIER to multiple watersheds individually before mosaicking the results together. The study area was subdivided into 46 smaller watersheds using HUC8 watershed boundaries (see Fig. 4.2 for geographical representation of the watersheds). This comparison is meant to assess whether dividing the region to run FIER individually and then mosaicking together impacts the accuracy of the flood predictions. The basins were selected by identifying watersheds that intersected the baseline area using a 50km buffer which is the maximum buffer size tested. This insured that we could compare the mosaicked outputs to the baseline area with every watershed that would have data from the baseline area while addressing the need to test applying FIER over larger areas. Given that the FIER method is data-driven, the patterns it can extract and the results of flood inundation are based on the data inputs. This can cause edge effects along watershed boundaries when mosaicking the individual FIER implementations. To mitigate potential edge effects and ensure smooth transitions between mosaicked watershed predictions, varying buffer sizes (0, 1, 2, 5, 10, 20, and 50 km) were tested when processing individual watersheds. The buffer sizes do not represent any relationship to the watershed size and are meant to represent a range from no buffer to a substantial one to identify general trends in applying buffer sizes to reduce FIER mosaicking edge effects.

The FIER framework requires that all pixels in a time-series be present to use to extract patterns, therefore only imagery with 99.9% clear sky conditions, a threshold to retain mostly clear scenes while allowing for minor imperfections in automated cloud masking. were used to limit the gaps in space for the extracted spatial patterns. Water fractions were predicted and evaluated using dates where the observed imagery had a greater than 90% clear sky conditions. The dates used for training were not evaluated for performance and used only to fit the FIER framework. This selection process retained approximately 5.3-21.6% of daily images for training and 25.1-40% for the evaluation dataset, varying by watershed due to cloud cover prevalence.

FIER predictions use truncated information from the modes that are correlated with hydrologic variables, meaning mathematically the predicted water fraction cannot maintain its original scale of 0 to 100%. Other research (Rostami et al, 2024) have applied a quantile mapping method to the FIER predictions which restore the complete signal range as much as possible. Quantile mapping is widely used in climate and hydrology studies to correct the biases in model-estimated values (Enayati et al., 2020; Farmer et al., 2018). The method matches the quantiles of Cumulative Distribution Functions (CDFs) from the predicted to the observed. The CDFs were calculated for all FIER trials (baseline and different buffers) using only the dates that were used for training on a per-pixel basis. The quantile mapping post-processing was applied for the prediction dates for all FIER trials. The two versions of FIER outputs, the original FIER predictions and post-processed predictions utilizing quantile mapping, were evaluated. This allows for evaluating the impact of post-processing on the accuracy of the mosaicked results compared to the baseline.

## 3.4 Statistical Analysis

To effectively compare the different experiments with the baseline, a statistical analysis was performed to understand and compare the accuracies as well as test if there are statistical differences between the experiments and baseline. The accuracy assessment employs multiple metrics: Structural Similarity Index Measure (SSIM) (Wang et al., 2004) to assess the spatial accuracy of flood extents, Root Mean Square Error (RMSE) to quantify the accuracy of intensity of flood predictions, Relative RMSE to assess the errors relative to the observed value, and lastly Mean Absolute Error (MAE) as another statistical measure to quantify the accuracy of intensity of flood predictions (Jackson et al., 2019).

The SSIM metric is defined by equation 1:

$$SSIM(x,y) = \frac{(2\mu_x\mu_y + C_1)(2\sigma_{xy} + C_2)}{(\mu_x^2 + \mu_y^2 + C_1)(\sigma_x^2 + \sigma_y^2 + C_2)}, \tag{1}$$

Where $\mu_x$ is the pixel sample mean of $x$, $\mu_y$ is the pixel sample mean of $y$, $\sigma_x^2$ is the variance of $x$, $\sigma_y^2$ the variance of y, $\sigma_{xy}$ is the covariance of $x$ and $y$, $x$ is the test image, $y$ is the reference image, $c_1 = (k_1 L)^2$, $c_2 = (k_2 L)^2$ two variables to stabilize the division with weak denominator, $L$ is the dynamic range of the pixel-values (in this case 100 to represent the range of water fraction), $k1 = 0.01$ and $k2 = 0.03$ by default. The SSIM index is calculated on various windows of an image, in this case we used an 11x11 Gaussian kernel for the calculation, and then averaged across the image to get the final SSIM metric. The SSIM has a range of -1 to 1 where 1 indicates perfect similarity, 0 indicates no similarity, and -1 indicates perfect anti-correlation.

The RMSE metric is calculated using equation 2:

$$RMSE = \sqrt{\frac{1}{n}\sum_i^n (x_i - y_i)^2}, \tag{2}$$

Where $x_i$ are the observations, $y_i$ are the observed values, and n is the number of observations.

The RRMSE metric is calculated using equation 3:

$$RRMSE = \sqrt{\frac{\frac{1}{n}\sum_i^n (x_i - y_i)^2}{\frac{1}{n}\sum_i^n (y_i^2)}}, \tag{3}$$

Lastly, the MAE is defined as equation 4:

$$MAE = \frac{1}{n}\sum_i^n |x_i - y_i|, \tag{4}$$

These error metrics were calculated for each pixel comparing the predictions to the observed and then averaged across the baseline area for each date of prediction. The baseline area was used to calculate the metric averages to keep the area consistent between the baseline and mosaicked results so that the results can be compared without influence of different areas.

The last statistical test used was the Kolmogorov-Smirnov test (Massey, 1951) to statistically compare the distributions of the evaluation metrics between the baseline and mosaicked results for both original and post-processed FIER outputs. This test compares whether two samples came from the same distribution. The test was performed for each metric and for every buffer size but keeping the baseline consistent. Lastly, the one-sided test was used to identify whether a given error metric was statistically greater than or less than the baseline. For the SSIM metric we tested if the mosaicked predictions are

significantly greater than the baseline. For the other error metrics (RSME, RRMSE, MAE) we tested if the mosaicked predictions are significantly lower than the baseline.

## 3.5 Case Studies

The statistical analysis described in the previous section was done using retrospective NWM streamflow as inputs into FIER, however, running FIER for actual flood extents will involve using the operational NWM streamflow predictions. Using case

studies serves to provide an evaluation of using the operational NWM for specific flood and low flow cases. We selected the statistically best FIER experiment for running the use cases.

To identify dates to use as cases for the low flow example, we selected a representative reach within the region close to the center of the baseline region along the Mississippi River. The streamflow data for the representative reach was averaged by month. The months with the lowest streamflow for 2019 and 2020 was used to select two dates (one from 2019 and one from

290 2020) as the low flow cases. These years were selected because they included operational NWM forecasts starting in late 2018 and overlap with available VIIRS imagery which had a substantial gap in data from 2021-01-01 to 2023-08-10. This process was done using the NWM operational analysis and assimilation data. Figure 3 displays the hydrographs for the reach where the low flow periods can be seen in 2019 and 2020. It should be noted that while the Mississippi River has been reported that increase in streamflow are expected due to long-term shifts in the climate Yin et al. (2023), the Mississippi

River has experienced droughts reported by recent research (Muñoz et al., 2023) leading to the downward trend in streamflow.

For the high flows, there were fewer options to select therefore a different approach was taken. We calculated the return periods for floods based on the NWM retrospective data for the representative reach within the region. We calculated the return periods on data from 1980-2018 using the Gumbel Type 1 distribution:

$$Q_{rp} = -\log\left(-\log\left(1 - \frac{1}{rp}\right)\right) \cdot \sigma \cdot 0.7797 \cdot \mu - (0.45 \cdot \sigma),$$ (5)

where $Q_{rp}$ is the return period flow, $rp$ is the return period in years, $\sigma$ is the standard deviation of the dataset, and $\mu$ is the average of the dataset. Next, we identified flood events from the NWM operational analysis and assimilation dataset from late 2018 - 2020 by comparing their peak flows to these calculated return period thresholds. We used the NWM operational product to determine flood events for the case studies because these are the data that would hypothetically be used in actual

situations for forecasting FIER. Another criterion for selection was finding dates where the data were reserved and not used

for training the models but used for evaluation. We selected a date that exceeded a 50-year return period in 2019 as well as a separate date that exceeded a 5-year return period as flooding case studies to evaluate FIER predictions. We selected these events in order to have an extreme flooding event and a more common flooding case.

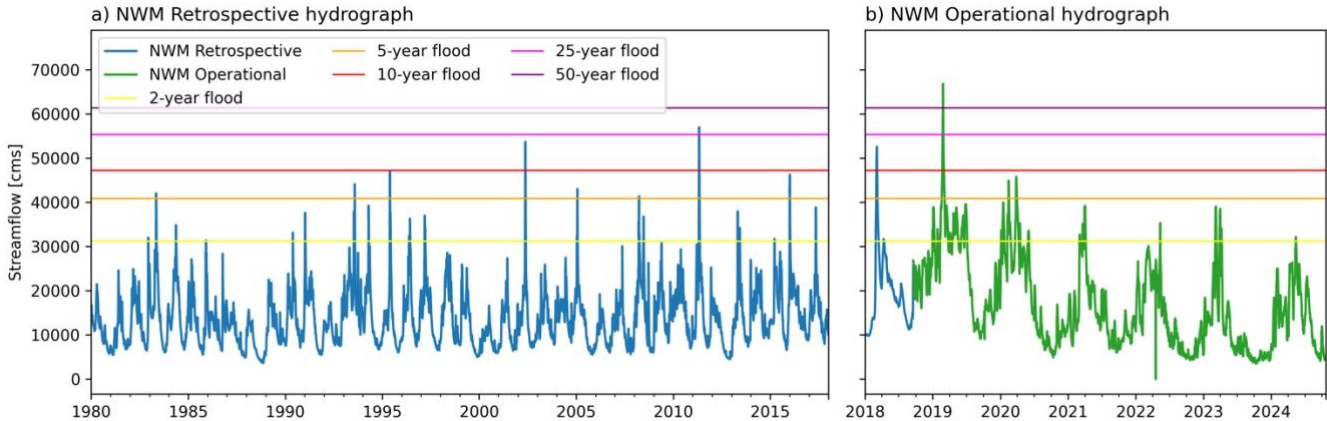

Figure 3: Streamflow hydrograph from the National Water Model (NWM) for the representative reach showing (a) retrospective simulations from 1980-2017 and (b) operational simulations from the analysis_assim run from 2018-2024. Horizontal lines indicate the 2-, 5-, 10-, 25-, and 50-year flood recurrence interval discharge values for the stream reach.

We ran FIER with streamflow data from a nowcast from the analysis and assimilation, 7-day medium-range forecast, and with a 15-day long-range forecast from the NWM to evaluate how different forecast runs and time horizons affect the FIER outputs. Since NWM produces sub-daily streamflow predictions, we averaged the streamflow values for the prediction date to use as inputs into FIER. For this initial scaling study daily average streamflow values were used as inputs to FIER to align with the daily VIIRS observation frequency and simplify the analysis, however, since NWM provides sub-daily streamflow the FIER predictions could be produced on a sub-daily time step and is a potential area for future refinement and operational use. The medium-range and long-range NWM streamflow predictions have multiple ensemble runs, so we averaged these across the different ensembles. Similarly, while NWM forecasts include ensembles, these were averaged for this study to provide a deterministic input to FIER. Incorporating ensemble streamflow to generate probabilistic inundation forecasts with FIER is a relevant avenue for future work. The medium-range and long-range streamflow predictions have multiple initialization times; we used the 00Z initialization for the forecasts to have a single forecast when calculating the streamflow values for the FIER extent predictions. We used the same statistics to evaluate the outputs from FIER for each of these use cases.

# 4 Results

## 4.1 Statistical Analysis

The experimental design aimed to assess the feasibility of applying the FIER method over larger geographic scales by segmenting the area of interest (AOI) into multiple smaller watersheds and then mosaicking the results together. An analysis of the REOF process and regressions are provided in Appendix A. Here we focus on comparing the results of the mosaicked process to running FIER over a larger baseline area. Figure 4 displays the average error metrics and how they vary with buffer sizes. First, applying FIER to multiple watersheds and then mosaicking the results does not lead to poor statistical performance compared to running FIER over the baseline AOI. When considering the original FIER outputs (green lines), Fig. 4 shows that the performance of the mosaicked results perform better than the baseline when the buffer size is smaller (0-10 km). When the buffer size is larger (20-50 km), the error metrics of the mosaicked results (green lines) trend more closely aligned with the baseline (dashed green line). For the SSIM metric specifically, the original mosaicked outputs at smaller buffer sizes (0-10 km) are not substantially greater than the baseline SSIM. For the RRMSE metric, the mosaicked results show higher errors from the original FIER outputs (no correction applied) compared to the baseline outputs but are closer to the baseline for the 1 and 2 km buffer experiments. When considering the post-processed outputs (blue lines), the mosaicked results perform better than the baseline for all the error metrics except for RRMSE. This is particularly notable for the SSIM metric, indicating that the post-processing applied to the mosaicked outputs are able to better capture the spatial patterns of observed flood inundation more effectively than either the baseline FIER output with post-processing or original outputs.

A noteworthy observation is the increase in error metric values for the post-processed FIER outputs compared to the original outputs, particularly noticeable in the RMSE and RRMSE values (Fig. 4 blue lines compared to green lines in graphs b and c). While there is also an increase in SSIM outputs with post-processing (Fig. 4a), the increase in SSIM shows improvement whereas an increase in RMSE/RRMSE indicates worse performance. Interestingly, the RMSE for the post-processed mosaicked results is much lower than that of the post-processed baseline output (blue lines in graph b). Moreover, this pattern of increasing error values is not present for MAE (graph d). RMSE and RRMSE gives more weight to larger errors, meaning these metrics are more sensitive to outliers than MAE. The results suggest that the post-processing reduces many smaller absolute errors but can introduce or amplify a few larger errors, particularly if the distributions of predicted and observed values used for CDF matching have significant differences in their tails or if the underlying flood patterns are complex and not perfectly captured by the limited modes.

Limited research has been conducted on properly validating inundation map and the interpretation. Those studies that have been published have focus on binary (water/no-water) maps (e.g. Schumann, 2019 and Landwehr et al., 2024). Conversely, there are well-known metrics and guidance for model for continuous variables from hydrologic simulations (e.g. Moriasi et al., 2005) however the guidance does not directly map to evaluating output images of continuous values such as water fraction. Therefore, the error metrics chosen and interpretation are based on important properties to evaluate for inundation

predictions (water fraction intensity and spatial patterns) and to compare different configurations without making universal claim of satisfactory outputs. Determining whether the outputs for a specific implementation of FIER should be the role of the practitioner considering their tolerance for specific errors.

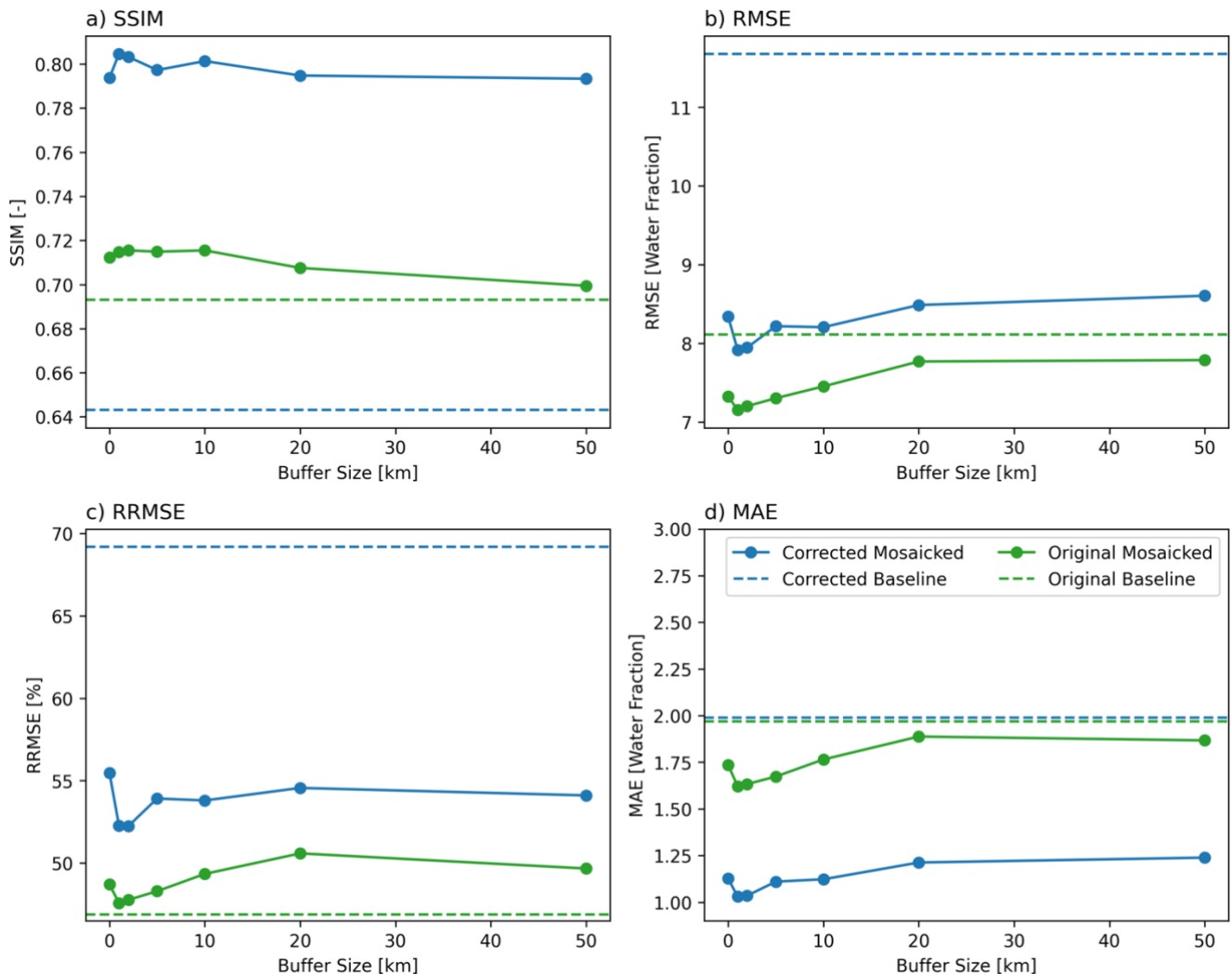

**Figure 4: Graphs illustrating the impact of varying buffer sizes on the performance of the FIER method for flood inundation mapping, assessed using the areal average of four different metrics: (a) Structural Similarity Index Measure (SSIM), (b) Root Mean Squared Error (RMSE), (c) Relative Root Mean Squared Error (RRMSE), and (d) Absolute Error. The green lines represent the original FIER outputs, while the blue lines depict the corrected outputs after applying a CDF matching post-processing step. The dashed lines indicate the baseline performance metrics obtained from applying FIER to a single, larger AOI.**

The other statistical analysis we performed was to compare the distributions of the errors. Figure 5 displays the CDF for the various error metrics and compares the baseline FIER predictions to the mosaicked FIER predictions with different buffer

sizes. The CDF plots show that the original outputs (top row) have similar curve shapes aside from at the upper quantiles (0.8 - 1) where the mosaicked results for SSIM metric show more values with better performance compared to the baseline (black line). Conversely, the RMSE, RRMSE, and MAE curves show more values with greater errors compared to the baseline. The plots displaying the post-processed CDFs (bottom row) display a different pattern compared to the original outputs. The SSIM curve for the mosaicked outputs displays better SSIM values compared to the baseline. However, the RMSE, RRMSE, and MAE CDF plots show that the post-processed mosaicked results have fewer values with less error (better performance) compared to the baseline for the lower quantiles (0 - 0.4) but have more values with higher errors (worse performance) at the upper quantiles (0.8 - 1) suggesting that the majority of errors for the post-processed results are due to large discrepancies with the observed. This supports the finding from comparing buffer sizes (Fig. 4) where the averaged RMSE metric worsens after the correction. The larger number of high error values can lead to higher RMSE and lower MAE errors because the RMSE metric is more sensitive to outliers than MAE.

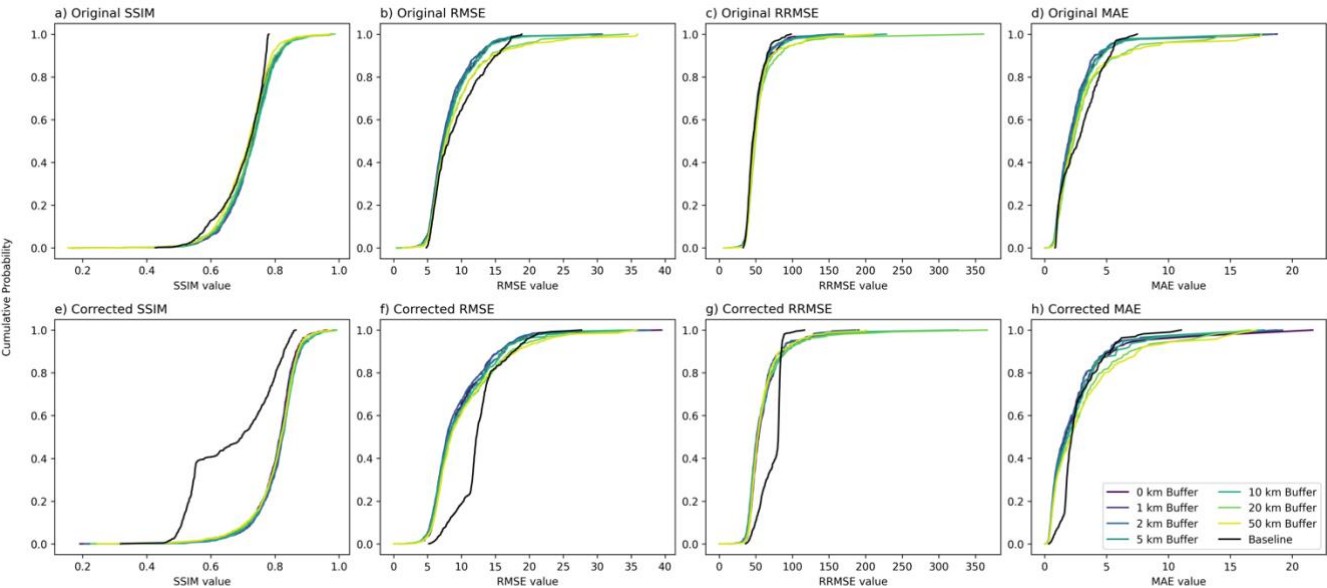

**Figure 5: Cumulative distributions of different error metrics for the FIER spatial scaling experiments. The top row (a-d) shows the original FIER outputs, while the bottom row (e-h) shows the outputs after applying CDF matching as post-processing. Different colors represent varying buffer sizes used when delineating individual watersheds for the mosaicked FIER approach. The black line represents the baseline FIER run over the larger area. The metrics include: (a, e) SSIM, (b, f) RMSE, (c, g) RRMSE, and (d, h) MAE.**

Table 1 provides the p-values from one-sided Kolmogorov-Smirnov tests, indicating whether the distributions from the various approaches are statistically different. The p-values from the Kolmogorov–Smirnov test reveal that there are significant differences between the baseline FIER and the mosaic approach across both the various error metrics and buffer sizes. In general, the buffer sizes of 20 and 50 km show that the mosaicked results are not statistically better (higher SSIM or lower RMSE, RRMSE, and MAE) than the baseline for the original outputs. Furthermore, the mosaicked results for buffer

sizes of 0-10 km have significantly better performance only for the SSIM and RMSE. While the mosaic outputs have values less than the baseline for the MAE metric, this difference is not significant. Additionally, the mosaicked outputs have a higher RRMSE than the baseline.

For the corrected FIER outputs, all buffer sizes show statistically significant differences in SSIM compared to the baseline with post-processing applied, indicating that by correcting the mosaicked outputs, the results are better able to capture the spatial distribution of flooding than using a single FIER process for a larger AOI. Furthermore, mosaic outputs show significantly better performance compared to the baseline AOI for the RMSE and MAE metrics. In particular, the p-values show that the buffer sizes 0-10 km are significantly less for RMSE, buffer size of 20 and 50 km are not significantly better. Whereas buffer sizes 0-20 km show statistically significant better performance compared to the baseline AOI for MAE and the buffer size of 50km is not significant. While the RRMSE metric was lower than the baseline for the corrected mosaicked results, the differences for all buffer sizes was not significant. This highlights that while post-processing aligns the spatial distribution of the flood predictions from FIER, it does not eliminate the inherent differences in the magnitude and spread of intensity-based errors between the two approaches. Overall, the Kolmogorov-Smirnov test results confirm that scaling FIER by mosaicking smaller watersheds produces statistically distinct error distributions. For the corrected outputs, the buffer sizes of 0-10 km generally result in more accurate predictions (higher SSIM, lower RMSE and MAE) compared to the original baseline outputs, and also often outperform the post-processed baseline outputs (as seen in Fig. 4).

Table 1: p-values from one-sided Kolmogorov-Smirnov tests comparing the distributions of evaluation metrics for FIER predictions with varying buffer sizes to the baseline FIER predictions. A lower p-value indicates a statistically significant difference between the distributions. Values denote significance at the 95% level are denoted with an asterisk (*), while values at the 99% level are denoted with double asterisks (**).

| Buffer | Original | | | | Corrected | | | |
|---|---|---|---|---|---|---|---|---|
| | SSIM | RMSE | RRMSE | MAE | SSIM | RMSE | RRMSE | MAE |
| 0 | 0.0419* | 0.0076** | 0.9999 | 0.9878 | 0.0076** | 0.0076** | 0.6925 | 0.0076** |
| 1 | 0.0241* | 0.0074** | 0.9813 | 0.1227 | 0.0074** | 0.0025** | 0.3697 | 0.0074** |
| 2 | 0.0239* | 0.0073** | 0.9948 | 0.1865 | 0.0073** | 0.0025** | 0.3658 | 0.0073** |
| 5 | 0.0234* | 0.0072** | 0.9973 | 0.1519 | 0.0072** | 0.0044** | 0.4869 | 0.0072** |
| 10 | 0.0219* | 0.0132* | 1.0000 | 0.9826 | 0.0067** | 0.0221** | 0.6211 | 0.0089** |
| 20 | 0.0876 | 0.1721 | 1.0000 | 0.9985 | 0.0064** | 0.0648 | 0.6122 | 0.0749** |
| 50 | 0.6267 | 0.2464 | 1.0000 | 0.9852 | 0.0121* | 0.1276 | 0.4941 | 0.1087 |

These findings suggest that scaling the FIER method spatially by mosaicking results from smaller watersheds is a viable approach based on the error metrics tested. The mosaicked approach consistently achieves comparable or better spatial pattern accuracy (SSIM) than the baseline. Furthermore, CDF matching generally improves the SSIM of the mosaicked

results and can improve RMSE/MAE compared to the original mosaicked outputs, leading to statistically significant improvements over the baseline, particularly for 0-10 km buffers. Furthermore, we found that the difference in mosaicked results error metrics are statistically significant. Larger buffer sizes (20 km and 50 km) do not show consistent improvements and may even lead to higher errors in some cases. This suggests that excessive buffering can blur the flood signal and reduce the accuracy of the predictions.

## 4.2 Case Studies

We performed the statistical analysis to understand the overall errors associated with running FIER over larger geographic scales using retrospective NWM streamflow as inputs, however, running FIER for actual flood extents necessitates using the operational NWM streamflow predictions for forecast predictions. We selected two dates with flooding and two dates with low flows then ran FIER for those time periods with only one buffer size to better understand how FIER does for extreme cases using the NWM operational data.

We selected which buffer size to use for the case study based on the statistical analysis that found a buffer size of 1 km or 2 km for mosaicked FIER outputs seems to be the most promising for operational use. While other buffer sizes show improvements in either SSIM or RMSE, we selected the 1 km buffer as it was found to strike a good balance between accurately capturing both the spatial extent and intensity of flooding when compared to the baseline. The corrected outputs for the 1km buffer generally demonstrate a better balance of error metric results (SSIM, RMSE, MAE) when considering overall performance improvements against the baseline and the impact of correction, which suggests that the CDF matching post-processing effectively reduces overall error in the flood inundation estimates, even if RMSE itself might see a slight increase for this specific buffer compared to its uncorrected version (Fig 4b). While we selected the 1km buffer size over a 2km buffer because the errors are slightly lower, these differences are marginal.

Figure 6 displays the results for running the mosaicked FIER process with the operational NWM predictions for the selected flood dates. Examining the 2019-02-25 flood event (top two rows), which exceeded the 50-year return period, FIER demonstrates consistent performance across nowcast, medium-range (7-day), and long-range (15-day) forecasts. The second row is zoomed into the baseline area to highlight more higher-resolution differences. Generally, it appears that the predictions capture the spatial dynamics of the flood, but the long-range prediction has a noticeably smaller extent than the nowcast or medium-range predictions. The 2020-02-15 flood event (bottom two rows), exceeding a 5-year return period, shows similar results where the forecasts are able to capture the extent of flooding, in this case, similar to the 2019 event's longer lead times, the further the lead time for prediction, the smaller the prediction of flood extents, suggesting more uncertainty with longer range forecasts. In particular, the long-range forecasts (Fig. 6, right column) also reveal complex spatial differences compared to the medium-range forecasts. For instance, while the overall extent might shrink with lead time, some localized areas, particularly in the eastern part of the region show decreased water fraction for the tributaries may possibly reflecting spatial variations in NWM forecast accuracy and its effects on the FIER outputs. Appendix B provides additional figures highlighting differences maps of the different forecasts compared to the observation. Table 2 tabulates the

error metrics for each simulation compared to the observation. While the long-range forecast exhibits slightly lower RMSE compared to the nowcast and medium-range for the 2019-02-25 flood event, the differences are marginal, and all forecasts achieve SSIM values above 0.57, indicating reasonable agreement with the observed flood extent. This suggests that FIER can provide reasonably consistent flood inundation predictions even with extended lead times, allowing for proactive flood mitigation measures.

The 2020-02-15 flood event, exceeding the 5-year return period, showcases comparable performance across forecasts horizons. The long-range forecast for this event achieves the highest SSIM of 0.67 and lowest RMSE of 16.04 among its predictions, highlighting FIER's potential for capturing more frequent flood events with reasonable accuracy. The nowcast for this event, however, shows a slight decrease in performance compared to the forecasts. The degradation in performance for the forecasts compared to the retrospective FIER simulations is likely due to the errors in the NWM streamflow predictions over the extended lead times.

**Table 2: Performance statistics for the FIER-predicted water fraction maps for two flood events 2019-02-25, the 50-year flood, and 2020-02-15, the 5-year flood.**

| Date | Forecast | SSIM [-] | RMSE [Water fraction] | RRMSE [%] | MAE [Water fraction] |
|---|---|---|---|---|---|
| 2019-02-25 | Nowcast | 0.5723 | 16.9233 | 78.482 | 3.9396 |
| | Medium range | 0.5766 | 17.2939 | 80.201 | 4.0435 |
| | Long range | 0.6664 | 16.2569 | 75.391 | 3.5162 |
| 2020-02-15 | Nowcast | 0.5414 | 18.4803 | 79.022 | 4.6667 |
| | Medium range | 0.6233 | 16.3844 | 70.060 | 3.6701 |
| | Long range | 0.6676 | 16.0370 | 68.575 | 3.4319 |

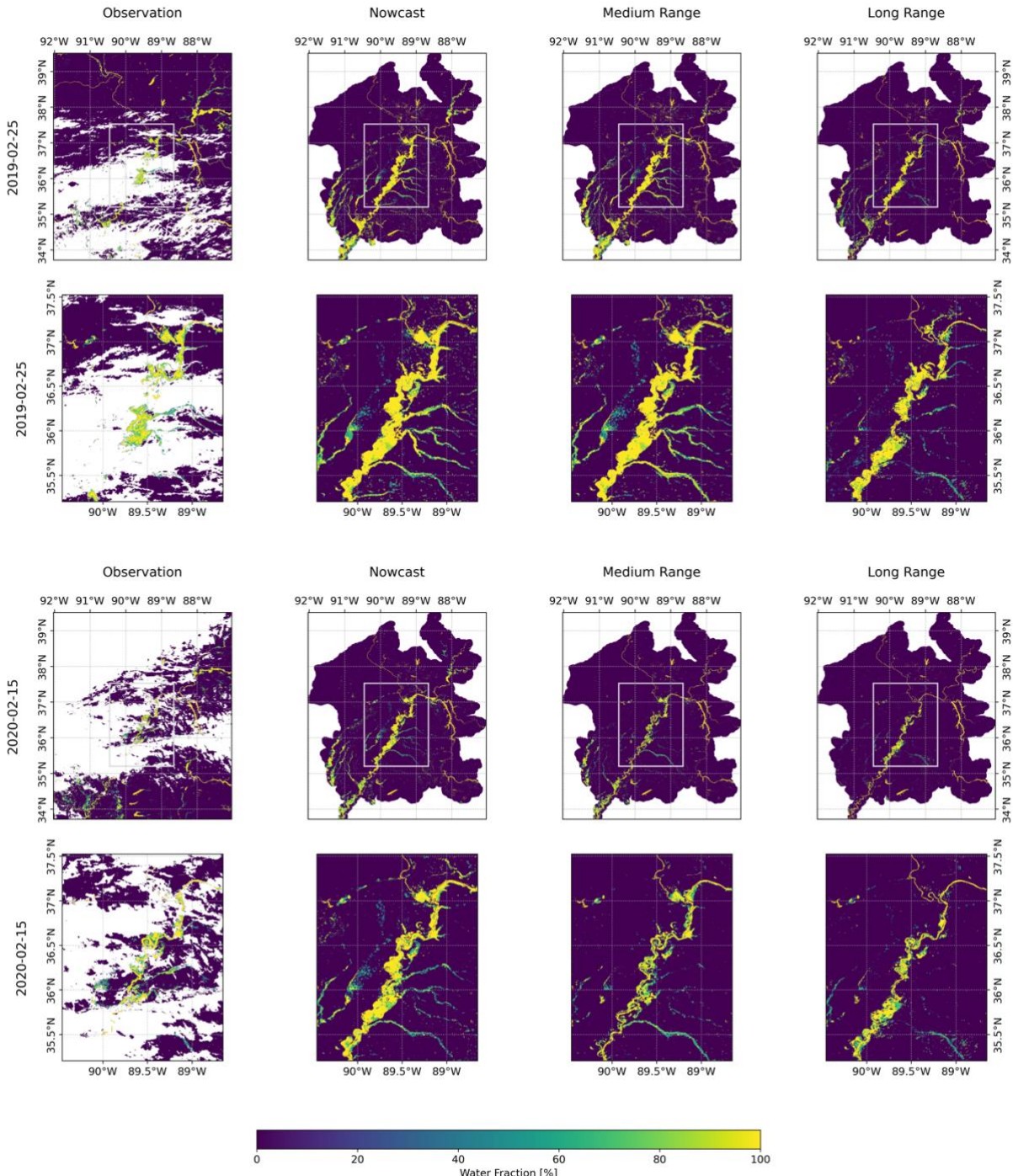

**Figure 6. Comparison of observed and FIER-predicted water fraction maps for the two selected flood events: 2019-02-25, the 50-year flood (top two rows), and 2020-02-15, the 5-year flood (bottom two rows) The lower rows are a zoomed view for the baseline area to highlight local differences in the predictions. Predictions use the nowcast, medium-range (7-day lead time) and long-range (15-day lead time) streamflow from the NWM. White areas are missing data due to clouds, cloud shadows and other poor-quality**
**data.**

The evaluation of FIER performance during low flow periods, using operational NWM streamflow predictions and the selected 1 km buffered and corrected configuration, reveals consistent and accurate results. Figure 7 displays the results for running the mosaicked FIER process with the operational NWM predictions for the selected low-flow dates. Table 3 shows the resulting error statistics comparing the FIER mosaicked predictions with the observation for the two selected low-flow dates. For the 2019-09-29 low flow event, FIER exhibits high SSIM values (above 0.73) across all forecast ranges, indicating strong agreement with the observed low flow conditions. The long-range (15-day) forecast demonstrates the lowest RMSE (8.53), RRMSE (59.00%), and MAE (0.89), suggesting that FIER can effectively capture low flow dynamics especially with extended lead times. This capability is particularly valuable for water resource management applications, such as drought monitoring and water allocation planning. Similarly, the 2020-10-21 low flow event shows consistent performance across all forecast ranges, with SSIM values exceeding 0.73. Again, the long-range forecast achieves the lowest RMSE (8.01) and MAE (0.79), reinforcing the ability of FIER to accurately predict low flow conditions with extended lead times. The visual comparison of the FIER-predicted water fraction maps (Fig. 7) with the observed data further supports these findings. The FIER outputs using the operational NWM streamflow outputs, while still capturing the general low flow patterns, show some minor deviations such as on the eastern side of the study regions for the 2019-09-25, likely attributed to the inherent uncertainties in the model streamflow predictions.

These results highlight the potential of the scaled FIER method for operational inundation prediction during low flow and its ability to generate reliable and accurate predictions of surface water extent during low-flow conditions,. This is important because an operational system should ideally perform sensibly whether flows are high or low. The consistent performance across different forecast ranges, coupled with the high SSIM values and low error metrics, demonstrates the capability of FIER to provide reliable and accurate low flow predictions. This information, by providing spatially explicit maps of surface water extent (e.g., surface water dynamics), can be crucial for informing water management decisions, such as simulating impacts due to dam constructions (Do et al., 2025).

**Table 3: Performance statistics for the FIER-predicted water fraction maps for two selected low-flow dates: 2019-09-29 and 2020-10-21.**

| Date | Forecast | SSIM [-] | RMSE [Water fraction] | RRMSE [%] | MAE [Water fraction] |
|------|----------|----------|-----------------------|-----------|----------------------|
| 2019-09-29 | Nowcast | 0.7381 | 9.8846 | 68.336 | 1.2012 |
| | Medium range | 0.7580 | 8.6568 | 59.848 | 0.9179 |
| | Long range | 0.7688 | 8.5340 | 58.998 | 0.8879 |
| 2020-10-21 | Nowcast | 0.7374 | 8.7395 | 61.268 | 0.9843 |
| | Medium range | 0.7394 | 8.7107 | 61.067 | 0.9571 |
| | Long range | 0.7713 | 8.0128 | 56.174 | 0.7921 |

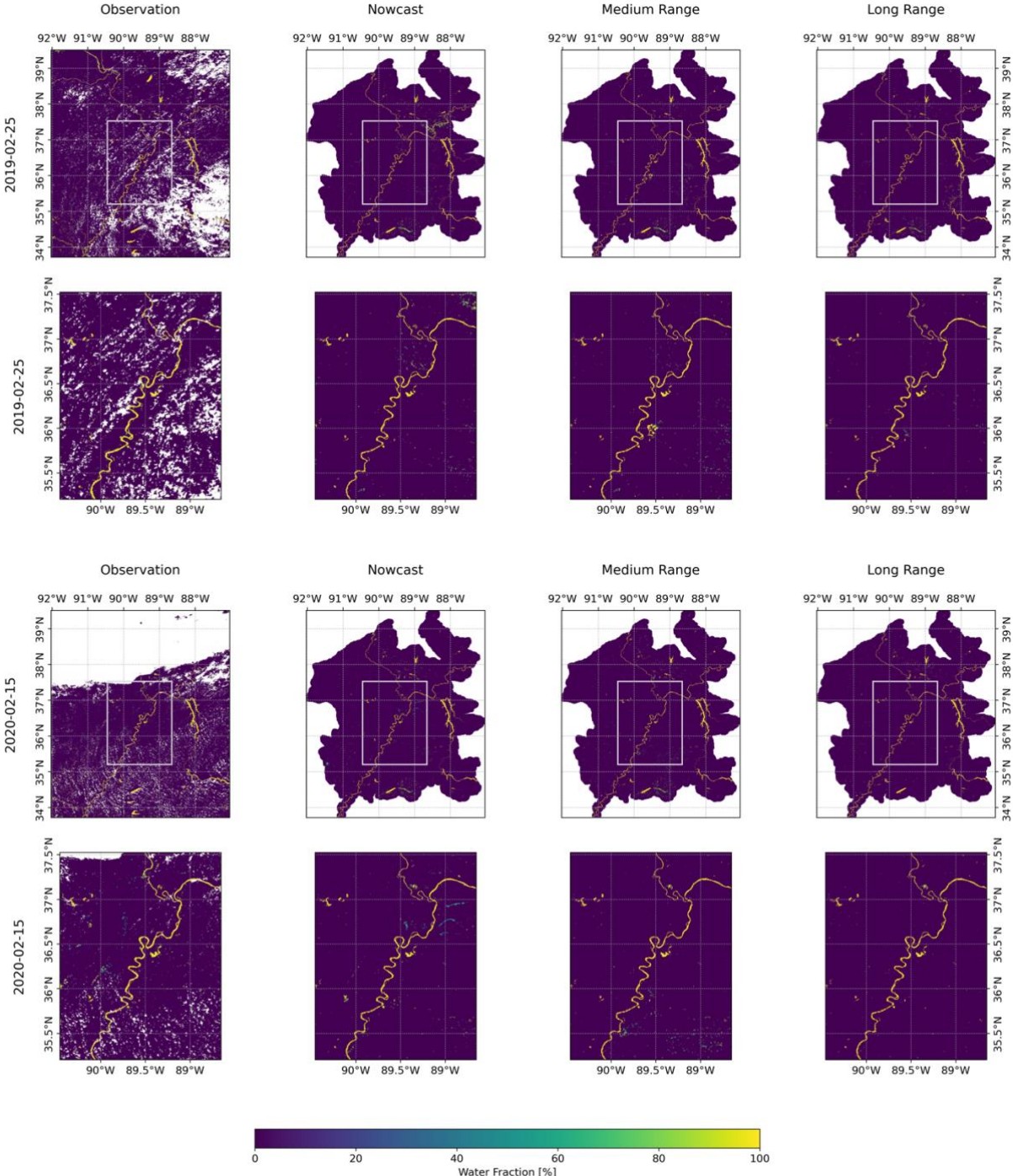

**Figure 7. Same as Fig. 6 except for the two selected low-flow dates: (top two rows) 2019-09-29 and (bottom two rows) 2020-10-21. Predictions use the nowcast, medium-range (7-day lead time) and long-range (15-day lead time) streamflow from the National Water Model. White areas are missing data.**

## 5 Discussion

### 5.1 Advantages and Limitations of the Watershed-Based Approach

Scaling the FIER method to larger geographic extents necessitates breaking the larger AOI into subunits. Hydrological regimes, topography, and flood characteristics vary significantly across different geographical locations, requiring regionally tailored implementations for accurate predictions. Furthermore, FIER is a data driven method meaning that the method is dependent on the data inputs and the patterns it can extract from the data. Using watersheds as the fundamental unit offers several advantages. Primarily, watersheds inherently delineate areas with interconnected hydrological regimes, ensuring that flood signals within each unit are driven by a common set of forcing factors. This allows for the development of regionally tailored FIER models that better capture the unique flood characteristics of each watershed. Additionally, by dividing a large area into smaller watersheds, the computational burden of FIER can be significantly reduced, facilitating parallel processing and enabling the application over extensive regions where limitations of computer resources can inhibit applying one FIER model.

Despite these advantages, the watershed-based approach presents certain limitations. One challenge lies in the potential for discontinuities at watershed boundaries when mosaicking individual FIER predictions. Abrupt transitions in predicted water fractions can arise due to variations in model parameters or data availability across watersheds. We showed that implementing the buffered approach around each watershed during the FIER fitting process allows for the mosaicking of multiple smaller FIER predictions to a large-scale surface water predict with reasonable accuracy compared to a baseline (Fig. 4) by incorporating information from neighboring areas and smoothing the flood signal across boundaries. However, the optimal buffer size is likely to vary depending on watershed characteristics and requires careful consideration. We found that selecting an excessively large buffer size risks blurring the flood signal and reducing the accuracy of predictions. More investigation is needed into what factors contribute to varying accuracy, particularly watershed characteristics such as size, topography, land cover, meteorology/climatology and flooding events.

Furthermore, the choice of watershed scale (i.e. HUC8 vs HUC12) has the potential to influence the performance of the scaled FIER model. Utilizing smaller, more numerous watersheds allows for finer details and can potentially capture localized flood dynamics more effectively. However, using more watersheds comes at the cost of increased computational complexity and the potential for greater boundary discontinuities. Conversely, larger watersheds simplify the mosaicking process but may omit fine-grained flood signals and fail to capture fine-scale variations in inundation extent. Ultimately, the optimal watershed scale and buffer size are likely to be site-specific and require careful evaluation based on the hydrological characteristics, computational resources, and desired level of spatial detail for the application. For this study we only used the HUC8 watershed scale for comparing how FIER performs when mosaicking; using different watersheds scales was out of scope for this work but is a topic for future research.

## 5.2 Implications for Large-Scale Flood Inundation Forecasting

The results from this study show that FIER can be successfully implemented over large areas using a mosaicking approach. The successful scaling of the FIER method holds significant implications for operational flood inundation forecasting at regional and continental scales. FIER's data-driven nature and computational efficiency make it particularly well-suited for large-scale applications where traditional hydrodynamic models are often computationally prohibitive or require extensive data inputs. By leveraging readily available satellite imagery and streamflow forecasts from hydrological models, like the NWM, FIER can provide rapid and accurate flood inundation predictions without the need for a complex modeling framework, high-resolution topographic data, or traditional calibration of physical parameters although the selection of optimal watershed scale and buffer size requires careful evaluation, akin to sensitivity analysis. This independence from detailed site-specific data typically required for hydrodynamic models such as high-resolution bathymetry and spatially distributed friction coefficients makes FIER a powerful tool for forecasting floods in data-scarce regions or ungauged basins, expanding the reach of flood inundation forecasting services to previously underserved regions (Chang et al., 2023; Do et al., 2025). It is important to note, however, that since FIER learns from historical data and the range of historical surface water dynamics, significant future alterations to floodplains such as new major flood control measures not present in the historical record or substantial shifts in hydrologic regimes could necessitate model retraining or adjustments to maintain forecast accuracy.

The scaled FIER method offers a valuable resource for a wide range of applications related to flood risk assessment, disaster preparedness, and water resource management. By providing timely and accurate flood inundation forecasts for events, FIER can support the development of effective early warning systems, enabling communities to prepare for and mitigate the impacts of flooding. In the context of water resource management, FIER can contribute to simulating flood responses to changes in hydrologic conditions which can inform flood risk assessments and guide long-term land-use planning decisions, optimizing reservoir building and operations (Do et al., 2025), assessing the effectiveness of flood control measures, and evaluating the impacts of climate change and human activities on flood regimes. Furthermore, FIER's ability to generate flood inundation maps from historical and even future long-term projected data can provide extents for specific return-periods which are vital with regards to climate change and planning (Wing et al., 2024). The scalability and computational efficiency of FIER hold promise to support large-scale flood inundation forecasting, enabling a more proactive and data-driven approach to flood risk management.

## 5.3 Caveats and Limitations

The watershed-based approach for scaling FIER, while promising, presents several limitations that warrant further investigation. The use of buffer zones, while mitigating abrupt transitions at watershed boundaries, may introduce artificial delineations that may not accurately represent the complex hydrological connectivity of real-world systems. For instance, in areas with complex topography or where floodwaters overtop watershed divides, the buffer zones may lead to inaccuracies in

570    the mosaicked flood predictions. As demonstrated in the statistical analysis, buffer sizes between 1-10 km showed the most promising results, but further research is needed to optimize buffer zone selection based on specific watershed characteristics and flood dynamics. Similarly, the choice of watershed scale presents a trade-off between spatial resolution and computational complexity. While smaller watersheds offer finer detail, they increase the potential for boundary discontinuities and computational burden. Conversely, larger watersheds simplify mosaicking but risks over smoothing the flood signal and missing localized flood events. The optimal scale is likely to be site-specific, requiring careful consideration of the desired level of detail and available computational resources.

Furthermore, the current FIER implementation's reliance solely on streamflow data as the hydrological driver, in this case NWM streamflow outputs. This limits its applicability in data-scarce regions or where streamflow observations are unreliable. Other large scale model data (e.g. GEOGLOWS, Hales et al., 2022) can be used and tested to understand how sensitive the approach is to input streamflow particularly for data-scare regions across the globe. GEOGLOWS provides 15-day streamflow predictions at 3 hour time steps for 51 ensemble streamflow predictions for approximately 7 million reaches across the globe, it is lower spatial resolution but has broader coverage as well as has different forecast horizons and ensemble members compared to NWM which would provide an interesting sensitivity analysis. Additionally, the focus on streamflow-driven flooding may not adequately represent other flood types, such as coastal flooding, flash floods, or pluvial flooding, which are not directly tied to streamflow variations. Expanding the framework to incorporate other hydrological variables, such as precipitation, soil moisture, and antecedent conditions, could enhance its robustness and broaden its applicability. It should be noted that the FIER processing framework as is can handle arbitrary hydrologic inputs for fitting but would require the user to prepare the data as inputs which is currently not implemented in the scripts shared. A critical aspect is the quality of the satellite-derived water fraction product used to train FIER, in this study the VIIRS data. While VIIRS provides valuable daily observations for large-scale monitoring, it is subject to inherent uncertainties beyond the already mentioned data gaps due to cloud cover . Integrating data from multiple satellite sensors, such as SAR and optical imagery could further improve the temporal density and quality of the input data (Markert et al., 2018). Furthermore, the accuracy of satellite-derived water fraction estimates can be significantly affected by land cover type; for instance, detecting surface water is notoriously challenging under dense forest canopies, within complex urban environments where water surfaces may be small or obscured, or in areas with extensive emergent vegetation (Li et al., 2020). Shadows and certain soil types can also be misclassified as water. Additionally, automated QAQC processing for satellites provide satisfactory cloud masking the masking trades off between over- and underestimation of clouds, resulting in misclassified clouds. These uncertainties in the input satellite imagery, representing potentially imperfect and incomplete inundation areas, directly propagate into the FIER model during its training phase, as the process extracts historical inundation patterns from this data. Consequently, FIER predictions may inherit these imperfections, potentially leading to underestimations of flood extent in areas where water is obscured by vegetation or overestimations due to misclassifications. This implies that for real-world hydrological applications, particularly those requiring high precision in specific challenging land cover types (e.g., detailed damage assessment in forested floodplains or urban flood mapping), the potential for such inaccuracies in FIER outputs must

be carefully considered. It is important to note that given FIER is a data-driven approach and determines flooding signals from historical data, there could be deviations from the data that are used to train the models and significant long-term changes in hydrological regimes due to climate change or floodplain characteristics due to new infrastructure or defenses. These changes could reduce the accuracy of FIER predictions for long-term operational use. However, periodic retraining should be completed to account for such changes caused by climate change or infrastructure. Finally, it is crucial to acknowledge that the performance of the scaled FIER model may vary across different geographic regions with diverse hydrological regimes and flood characteristics (Prince et al., 2025). The location used in this study along the Mississippi River is humid with varying land cover types compared to the Western US which is much drier but also experiences flooding where FIER may be applicable. Further validation and testing in various environments are necessary to assess its generalizability and transferability beyond the study areas examined in this research.

## 5.4 Future Work

While this study demonstrates the potential of the scaled FIER method for large-scale flood inundation forecasting, several avenues for future research can further enhance its accuracy and applicability. First, exploring alternative regression methodologies between streamflow and the FIER-derived temporal patterns (RTPCs) could improve the model's ability to capture complex flood dynamics. Other studies (e.g. Chang et al., 2023, Rostami, et al., 2024) have used dense neural networks (DNN) to create the forecasts in the FIER framework. Incorporating non-linear regression techniques, like DNN, in a scalable manner and additional hydrological variables, such as precipitation and soil moisture, may enhance the model's predictive capabilities. Second, addressing the limitations of input satellite data is crucial for enhancing FIER's reliability (Wan et al., 2025). Advanced data fusion methods, which synergistically combine optical data (like VIIRS) with SAR and potentially higher-resolution sensors, offer a promising avenue particularly with respect to gaps due to cloud cover (Markert et al., 2018, 2024). Such fusion approaches can provide more complete and accurate historical flood observations, thereby improving the quality of the data used to train FIER and, consequently, the reliability of its predictions. Further research could also focus on developing land cover specific error characterizations for input satellite data or incorporating ancillary datasets to characterize water detection accuracy. A quantitative assessment of NWM streamflow forecast errors and their direct propagation into FIER inundation uncertainty was beyond this study's scope but is crucial for future operational implementation. A systematic analysis of the relationship between buffer size and watershed characteristics is crucial for optimizing the mosaicking process including buffer sizes relative to watershed area. By examining factors like watershed size, shape, topography, and land cover, we can develop guidelines for selecting appropriate buffer sizes for different regions, minimizing boundary discontinuities while preserving the accuracy of individual FIER predictions. Additionally, it was mentioned that only one watershed scale was tested (HUC8) and different sizes of watersheds need to be tested to understand how the FIER results will be affected by watershed sizes. This can lead to a hybrid approach where buffer size and watershed scale for running FIER can be data-driven and yield better results over large areas. Finally, to further enhance computational efficiency, masking out watersheds with historically limited flooding from the analysis can significantly

reduce processing time. This targeted approach focuses computational resources on areas most prone to flooding, enabling more efficient application of FIER over large geographic extent.

## 6 Conclusion

This study aimed to address the critical need for efficient and accurate large-scale flood inundation forecasting by applying the FIER method, a data-driven technique previously demonstrated at smaller scales, over large areas. Recognizing the limitations of traditional hydrodynamic models and the need for a computationally efficient approach for operational flood forecasting, we investigated the feasibility of using a watershed-based approach to scale FIER, leveraging the inherent hydrological connectivity of watersheds and then mosaicking results to create a single flood map for a given simulation. Our

analysis focused on flood-prone regions in the United States, the Upper Mississippi Alluvial Plain, where flooding occurs often.

The results demonstrate the effectiveness of the watershed-based approach for scaling FIER. Statistical analysis of the mosaicked FIER predictions, using retrospective NWM streamflow data, revealed that buffer sizes of 1-10 km achieved the best balance between accurately capturing the spatial extent (SSIM) and intensity (RMSE) of flooding. The average SSIM

metric ranged from 0.714 to 0.715 for the original FIER outputs and 0.797 to 0.804 for the corrected outputs. The average RMSE metric ranged from 7.15 to 7.45 percent for the original FIER outputs, and from 7.91 to 8.21 percent for the corrected outputs. Notably, the corrected FIER outputs, using a CDF matching post-processing technique, consistently showed better SSIM error values compared to the original outputs but higher RMSE and lower MAE, suggesting the overall error was reduced but also introduced larger errors. Overall, the correction improves the predictions and yields significantly better

error metrics compared to a baseline for the 1-10 km buffer sizes. Case studies using operational NWM streamflow forecasts for specific flood and low flow events further validated the performance of the scaled FIER method. The 1 km buffered and corrected FIER outputs were used for the case study and coupled with NWM forecasts with varying forecast lead times. These flood extent predictions accurately forecasted both the extent of inundation, achieving SSIM values above 0.54-0.66 for flood events and above 0.73 for low flow events.

The watershed-based FIER approach offers several advantages, including the ability to capture regional flood characteristics and the ability to set up and run efficiently without the need of prohibitively expensive hardware resources. However, limitations such as boundary effects, sensitivity to watershed scale, and reliance on streamflow data require further investigation. Future research should focus on optimizing buffer zone selection based on watershed characteristics, exploring alternative regression methodologies, incorporating additional hydrological variables, and expanding the framework to

encompass non-fluvial flood processes. These advancements will further enhance the scalability, accuracy, and applicability of FIER for large-scale flood inundation forecasting, enabling more effective flood risk management and water resource planning.

**Appendix A: FIER fitting statistics**

We performed additional analysis to investigate and highlight the results of the fitting process between the REOF and hydrologic data. We separated the analysis into the baseline area and the full area with the different watersheds.

For the baseline area, we found that the first three modes of spatio-temporal patterns were significant and account for about 93% of the total variance of the VIIRS water fraction image time series. Hereafter, the first modes of the RSM or RTPC will be called RSM-01 or RTPC-01, respectively with the second mode of RSM or RTPC will be called RSM-02 or RTPC-02, respectively, and so forth.

Figure A1 displays the results of the REOF and fitting for the baseline area. The top row displays the first three RSMs, revealing distinct spatial flooding patterns captured by VIIRS water fraction data. RSM-01 exhibits a widespread pattern encompassing the main stem of the Mississippi River and its tributaries, suggesting a dominant mode of flooding associated with high flows in the main channel. RSM-02, highlights localized flooding patterns in the southeastern portion of the basin, potentially indicating areas susceptible to backwater effects or tributary flooding. RSM-03 shows a more dispersed pattern with both positive and negative values, suggesting a complex mode of flooding that may be influenced by a combination of factors. The middle row in Fig. A1 presents the time series of the corresponding RTPCs and normalized NWM streamflow for the representative reach. The close alignment between the RTPC fluctuations and streamflow variations, particularly for RTPC-02, indicates a strong correlation between these temporal patterns and the hydrological driver. The scatter plots (bottom row) show the relationship between RTPCs and normalized NWM streamflow. While linear regression was used here for simplicity and consistency with previous FIER implementations (Chang et al., 2020), some scatter plots (e.g., for RTPC-01) suggest that more complex, non-linear fitting functions or robust regression methods accounting for outliers might yield different fits. The influence of high-leverage points on the linear fit is apparent in some cases. Exploring alternative regression techniques such as those implemented by (Rostami et al., 2024) or (Wan et al., 2025) is a key area for future FIER development as discussed in Section 5.4). The three RTCPs have a Pearson's R correlation coefficient of 0.7342, 0.8722, and 0.7615 for modes RTPC-01, RTPC-02, and RTPC-03, respectively. This correlation between the RTCPs and streamflow is further quantified in the bottom row, which shows scatter plots of the RTPCs against normalized NWM streamflow, along with the fitted regression models. The Nash-Sutcliffe Efficiency (NSE) values (0.61, 0.77, and 0.63) for the three fitted models, generally considered to indicate acceptable to good performance in hydrological modeling, confirm the positive statistical relationships established between the flood patterns and streamflow, demonstrating the effectiveness of the regression models in capturing these relationships.

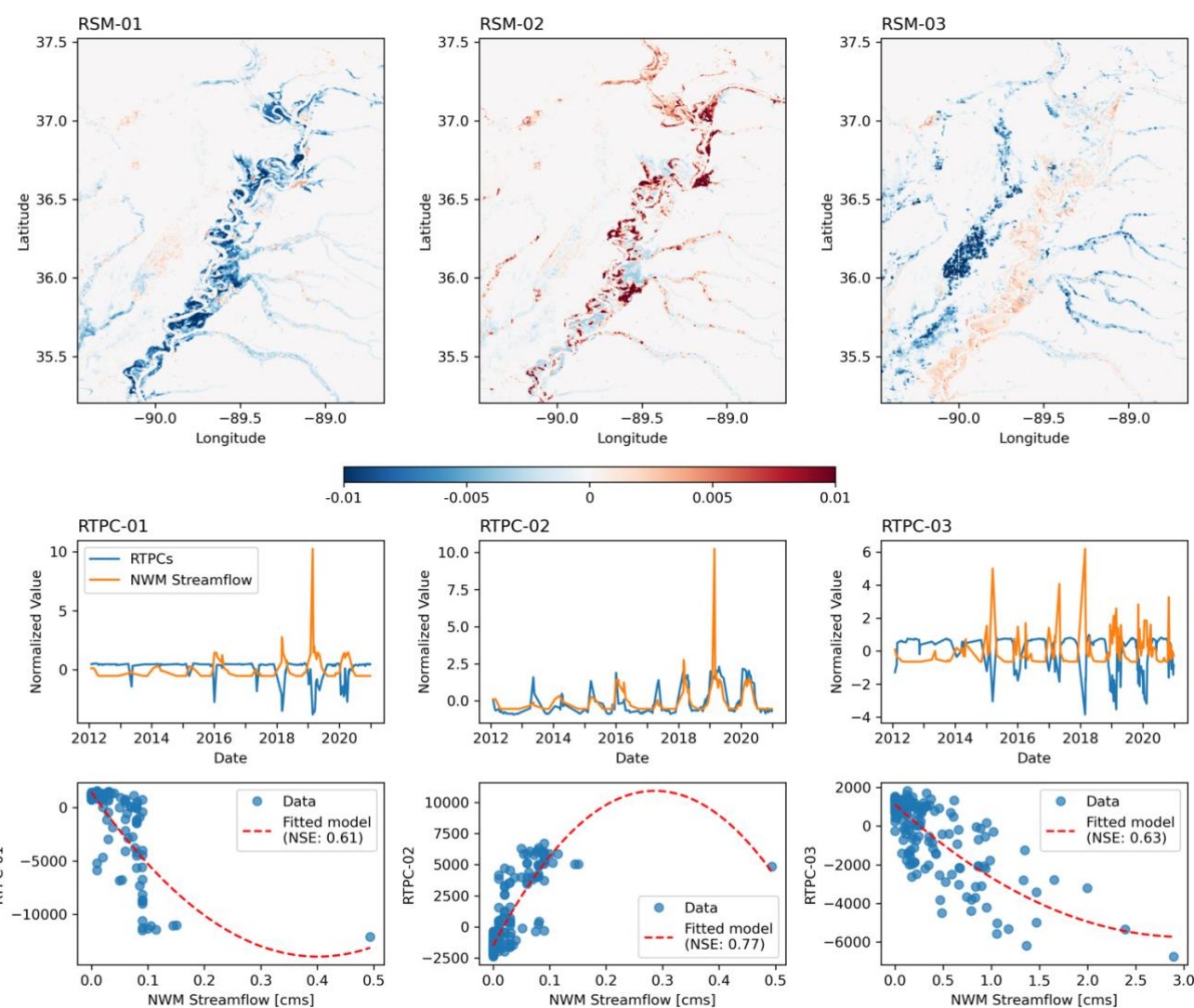

**Figure A1: Spatiotemporal patterns and regression models for the baseline area. (Top) The first three Rotated Spatial Modes (RSMs) derived from REOF analysis of VIIRS water fraction data. Red colors generally indicate positive correlations with the modes RTPC (increased water fraction when the mode is active), while blue colors indicate negative correlations or opposing RTPC. (Middle) Time series of the corresponding Rotated Temporal Principal Components (RTPCs) (blue) and normalized NWM streamflow (orange) for the representative reach. (Bottom) Scatter plots of the RTPCs against normalized NWM streamflow, along with the fitted regression models (red dashed lines) and their corresponding Nash-Sutcliffe Efficiency (NSE) values.**

Figure A2 provides a view of how the performance of the scaled FIER method varies across different buffer sizes and watersheds. Given that there is so much data across the different REOF and regression processes, Fig. A2 summarizes the number of RSMs (left column), correlation between RTCPs and streamflow (middle column) and NSE from the fitted model (right column) for each watershed. Table A1 also provides the mean and stand deviation in parenthesis for each of the

metrics across all watersheds. Examining the number of significant RSMs, we observe a general trend of an increasing number of significant RSMs with larger buffer sizes. This suggests that incorporating information from neighboring watersheds through buffering enhances the ability of REOF analysis to capture distinct flood patterns. However, the average Pearson's correlation coefficient between RTPCs and streamflow remains relatively consistent across buffer sizes, ranging from an average of 0.727 to 0.765. This indicates that the strength of the relationship between flood patterns and streamflow is not significantly affected by the buffer size. Interestingly, the average NSE of the fitted regression models shows a more nuanced pattern. While smaller buffer sizes (0-2 km) exhibit relatively lower NSE values, indicating moderate model performance, the NSE gradually increases with larger buffer sizes, peaking at 0.616 for the 50 km buffer. This suggests that incorporating broader spatial context through larger buffer zones can improve the predictive capability of the regression models. However, it's important to note that the standard deviation of NSE also varies with buffer sizes. Overall, the analysis suggests that while the number of significant modes and the strength of the correlation between flood patterns and streamflow are not significantly impacted by buffer size, larger buffer zones can potentially enhance the predictive accuracy of the regression models. However, the increased variability in model performance with larger buffer sizes necessitates a careful consideration of the trade-offs between model complexity and accuracy when selecting the optimal buffer size for a given application. It is important to distinguish these REOF component fitting statistics from the final inundation map accuracy metrics discussed in the main paper. While better component fits (e.g., higher NSE for the RTPC-streamflow regression) are generally desirable, the optimal buffer size for the final map accuracy (e.g., SSIM, RMSE) involves a balance, as very large buffers might improve individual component fits by incorporating more data but could also smooth out critical flood details or introduce noise from distant, less relevant areas, potentially degrading the final mosaicked inundation map.

**Table A1: Summary statistics of REOF analysis and regression model performance for varying buffer sizes. The table shows the range of significant modes, average Pearson's correlation coefficient, and average NSE for each buffer size, with standard deviations in parentheses.**

| Buffer size | Range of Significant RSMs | Avg Pearson's R | Avg fit NSE |
|---|---|---|---|
| 0 km | 1 - 8 | 0.727 (0.038) | 0.587 (0.060) |
| 1 km | 1 - 8 | 0.728 (0.039) | 0.586 (0.063) |
| 2 km | 1 - 8 | 0.732 (0.037) | 0.594 (0.070) |
| 5 km | 1 - 8 | 0.741 (0.041) | 0.611 (0.070) |
| 10 km | 1 - 8 | 0.742 (0.032) | 0.599 (0.052) |
| 20 km | 2 - 10 | 0.745 (0.037) | 0.602 (0.058) |
| 50 km | 1 - 11 | 0.765 (0.029) | 0.616 (0.049) |

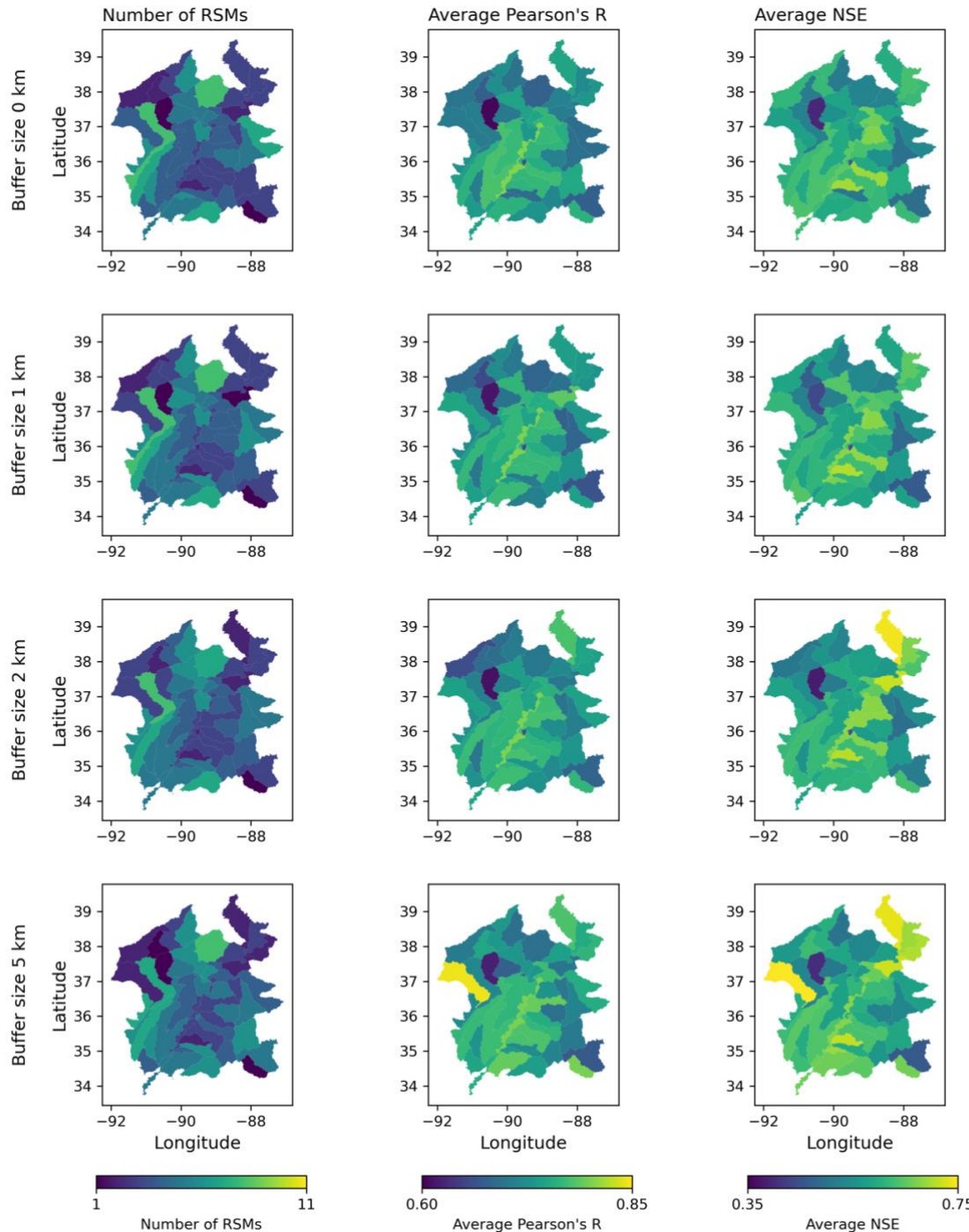

**Figure A2: Spatial distribution of FIER model performance metrics for varying buffer sizes. The figure displays maps showing (left) the number of significant RSMs identified by the Monte Carlo test, (middle) the average Pearson's correlation coefficient between the RTPCs and streamflow, and (right) the average Nash-Sutcliffe Efficiency (NSE) of the fitted regression models for each watershed, across different buffer sizes (0 km, 1 km, 2 km, 5 km).**

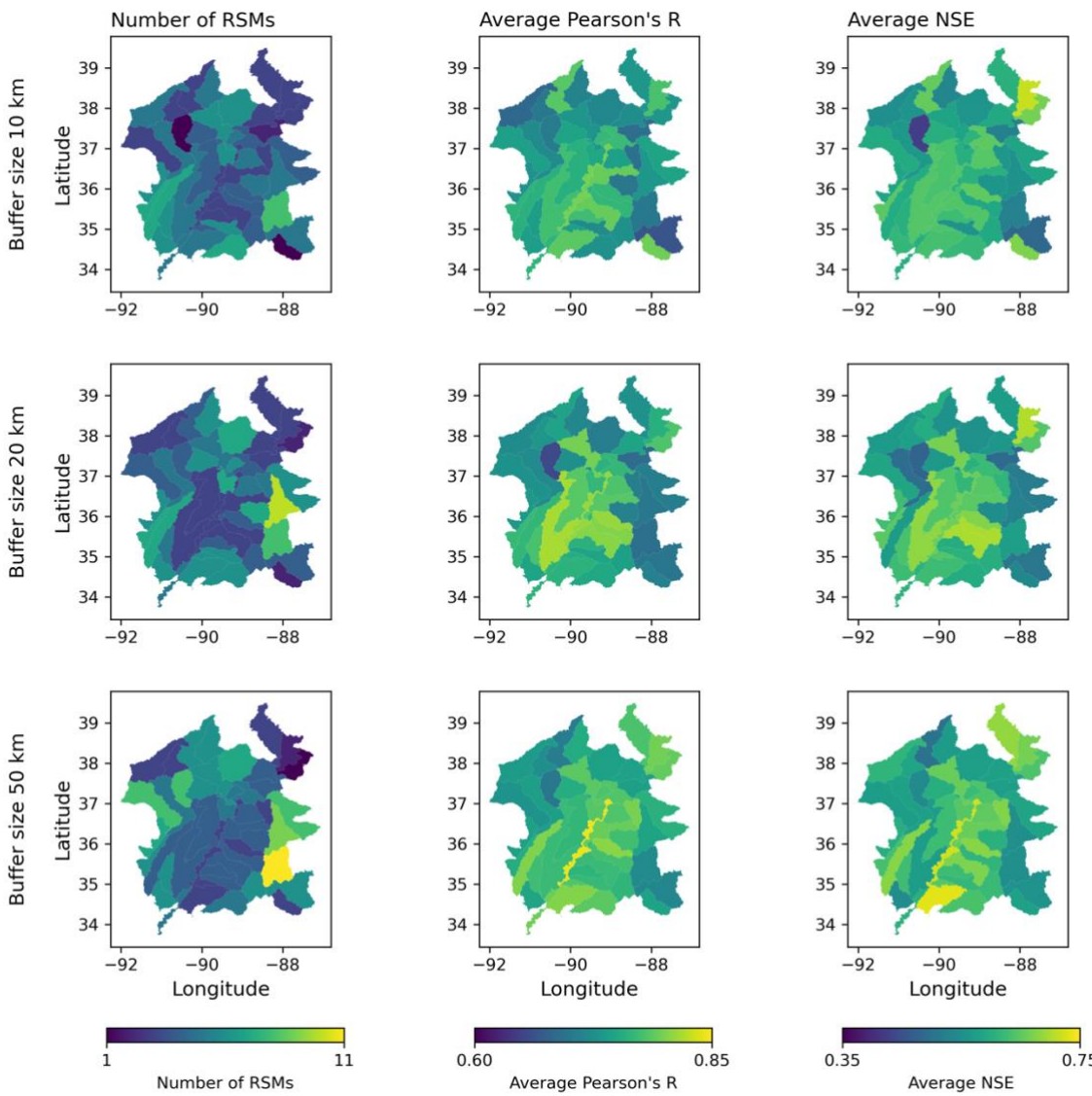

**Figure A2 (continued): Same as Fig. A2 but showing for the buffer sizes 10 km, 20 km, and 50 km**

## Appendix B: Case Study Difference Maps

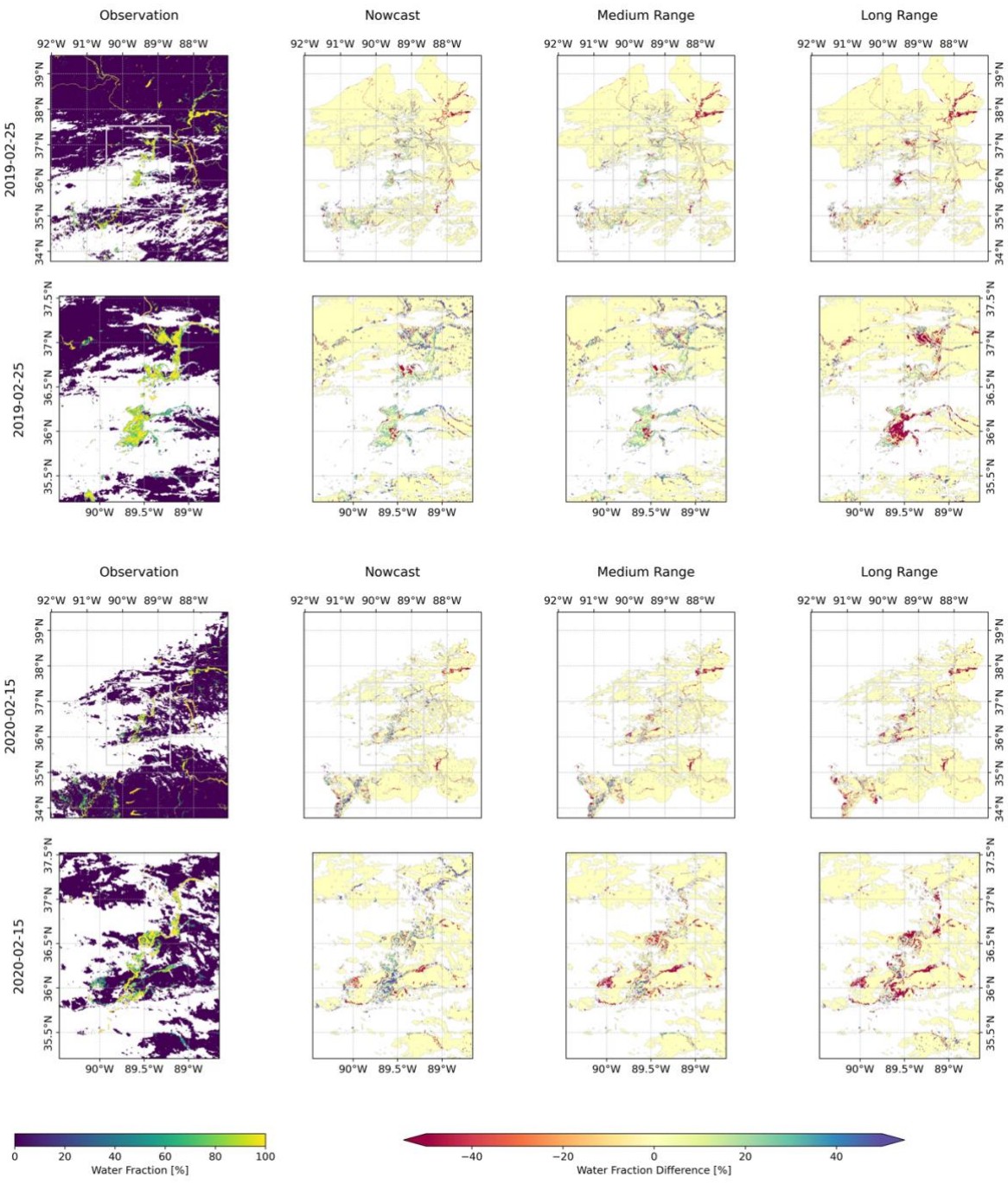

**Figure B1: Maps highlighting the observed water fraction from VIIRS and difference from FIER-predicted water fraction maps (predicted – observation) for the two selected flood events: 2019-02-25, the 50-year flood (top two rows), and 2020-02-15, the 5-year flood (bottom two rows). Warmer colors show underprediction from FIER whereas cooler colors show over prediction. White areas are missing data. To be used as comparison with Fig. 6**

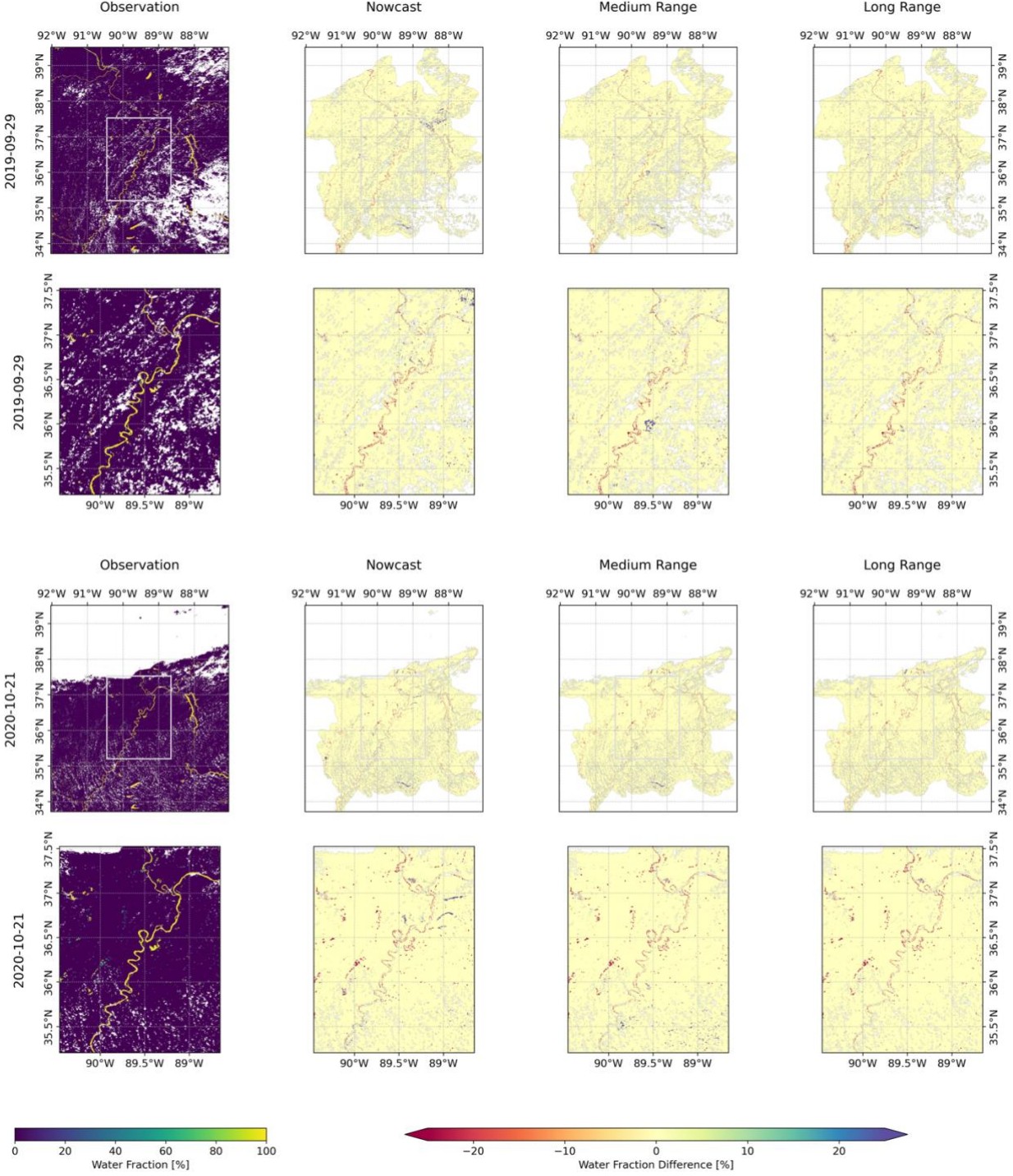

**Figure B2: Same as Fig. B1 except for the two selected low-flow dates: (top two rows) 2019-09-29 and (bottom two rows) 2020-10-21. To be used as comparison with Fig. 7**

### Code availability

The software/scripts used in this study for data processing, analysis and figure generation are publicly available under the open-source Apache 2.0 license. The source code used can be accessed at https://github.com/KMarkert/phd-fier-scaling (accessed on 7 Nov 2024). Developer: K.N.M.; year first available: 2024; license: Apache 2.0; programming language: Python.

### Data availability

The data used in the study are openly available online. The VIIRS water fraction data can be accessed from the AWS Registry of Open Data, specifically the NOAA Joint Polar Satellite System (JPSS) cloud storage bucket (https://registry.opendata.aws/noaa-jpss/). We processed and stored the VIIRS water fraction data for North America publicly on Earth Engine with the following collection ID: "projects/byu-hydroinformatics-gee/assets/noaa_jpss_floods", we make no explicit guarantees to maintaining the collection on Earth Engine and the NOAA source should be considered the authoritative source. The NWM retrospective data can be accessed through the AWS Registry of Open Data from the NOAA National Water Model CONUS Retrospective Dataset (https://registry.opendata.aws/nwm-archive/). The operational NWM data can be accessed via the Google Cloud Public Dataset on BigQuery (https://goo.gle/nwm-on-bq).

### Author contribution

Conceptualization, K.N.M., H.L, G.P.W., E.J.N. and D.P.A.; methodology, K.N.M.; software, K.N.M.; validation, K.N.M.; visualization, K.N.M., G.P.W, and H.L.; supervision: G.P.W., E.J.N., H.L., D.P.A. and R.E.G.; writing—original draft preparation, K.N.M., G.P.W., E.J.N., H.L. and D.P.A.; writing—review and editing, K.N.M., G.P.W., E.J.N., H.L., D.P.A. R.E.G. and F.J.M.; All authors have read and agreed to the published version of the manuscript.

### Competing interests

K.N.M. is employed by Google; the methods presented use generally available Google technologies. D.P.A. is a member of the editorial board of the Environmental Modelling & Software journal. The other authors declare no conflicts of interest.

## Acknowledgements

The authors would like to thank the data providers, particularly the National Oceanic and Atmospheric Administration (NOAA) Open Data Dissemination (NODD) Program for providing the JPSS VIIRS data and NWM data freely available for use. We would like to thank Google Open Datasets Program for publicly hosting the operational NWM on BigQuery and to the Google Earth Engine team for the use of the Earth Engine platform under the non-commercial terms of service for research. Lastly we would like to thank the reviewers for their insightful comments that improved the quality of the manuscript. Gemini, a generative AI tool developed by Google, was used during manuscript editing to help improve clarity and proper grammar. Funding for research was provided by the NOAA JPSS Program (Grant No. NA20NES4320003).

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
