# Peer review of "Evaluating the Feasibility of Scaling the FIER Framework for Large-Scale Flood Inundation Prediction"

_EGUsphere, 2024_

## Author Response (AR1)

REVIEWER 1 RESPONSE:

We sincerely thank you for your detailed and insightful review of our manuscript. Your comments are highly valuable and have helped us identify several areas where the clarity, justification, and overall impact of our work can be significantly improved. We appreciate your positive feedback on the relevance of the topic, the well-structured text, the open-source nature of our code, and the focus on operational applicability.

We have carefully considered all your general and detailed comments and have made corresponding revisions to the manuscript. Below, we address each of your points and describe the changes made.

**General Comments:**

1. **Abstract Clarity (Data-driven vs. data-scarce, metrics, buffer sizes):**
   ○ We agree that the distinction between "data-driven" and "data-scarce" could be clearer. We have revised the abstract to specify that FIER is data-driven with respect to historical satellite imagery and streamflow, but offers a solution in regions typically considered "data-scarce" for traditional hydrodynamic modeling (i.e., lacking detailed bathymetry, friction coefficients, etc.).
   ○ Regarding metrics and buffer sizes in the abstract: We have aimed for a balance. We've opted to remove the key metric ranges as they highlight the quantitative improvement but added a little more context with significance tests used and clarified the post-processing method. We have stated the need for the buffers in the abstract but feel that adding context for size of buffers would not add value so opted to keep as is. We believe this provides key takeaways with a high level overview without overly lengthening the abstract.

2. **Mixed Signals on Hydraulic/Hydrologic Models:**
   ○ Thank you for pointing this out. We have revised the introduction (Section 1) to acknowledge upfront there are errors with both hydrologic and hydrodynamic modeling. Furthermore we clarify that while FIER leverages hydrological model outputs (like NWM), these models also have their own uncertainties which can influence FIER outputs. We now clarify that FIER's advantage lies in *not requiring the user to develop and calibrate a complex hydrodynamic model from scratch for inundation mapping*, instead utilizing existing, often operational, streamflow forecasts.

3. **Impact of Climate Change/Infrastructure on a Data-Driven Method:**
   ○ This is a very important point. We have added a discussion in "Caveats and Limitations" (Section 5.3) acknowledging that FIER, being data-driven, learns from historical patterns. Significant long-term changes in hydrological regimes (due to climate change) or floodplain characteristics (due to new infrastructure or defenses) that deviate substantially from the training period could reduce the

accuracy of FIER predictions. We state that periodic retraining to account for such non-stationarity would be necessary.

4. **Study Area Description:**
   ○ We concur that a more detailed description would benefit readers. We have expanded Section 3.1 ("Study Area") to include more specifics on the hydrology of the study area, its main flood drivers (e.g., contributions from major tributaries like the Ohio, influence of snowmelt vs. rainfall), more information on infrastructure like reservoirs and levees, and the values of of upstream inflows. We added additional context on droughts to provide context and justification of the study area for low flow predictions.

5. **Explanation and Benchmarking of Error Metrics:**
   ○ We acknowledge the challenge of defining absolute benchmarks for flood inundation map accuracy, as these are often application-specific and not universally established. In the revised manuscript (primarily in Section 4.1 "Statistical Analysis" and the Discussion), we emphasize that our evaluation focuses on *relative performance* (watershed-based approach vs. baseline, impact of buffers, effect of post-processing) and consistency across different forecast lead times. We have added text to Section 4.1 to briefly contextualize why validating water fraction maps is difficult compared to previous research and evaluation guidance. The manuscript states the metrics like SSIM and RMSE/MAE signify in terms of capturing spatial patterns and water fraction intensity. While we cannot provide definitive "satisfactory thresholds," we put language in the text that practitioners should determine what is good enough for operational implementations..

6. **Low-Flow Use Case:**
   ○ We appreciate your perspective on the low-flow case. We agree its primary strength in this manuscript is not to position FIER as a drought forecasting tool, but rather to demonstrate its robustness for simulating water for all cases and operational use. We have revised the introduction to frame the objectives of general inundation mapping for both flood and low-flow conditions. We updated discussion (Section 5) to highlight that FIER's ability to *not* generate spurious flood signals during low-flow conditions, and to reasonably represent surface water extent, increases confidence in its operational reliability across a range of hydrological conditions. This is important because an operational system should ideally perform sensibly whether flows are high or low.
* * *
**Detailed Comments:**

● **[Line 29] Buffer values in abstract:** We have added context to the abstract when introducing the buffer approach.  Going into details in the abstract will make it too long so we opted to leave that for the text.
● **[Line 66] "indeed" a "promising avenue":** We have rephrased this to avoid the unsubstantiated "indeed" and be more clear about data inputs from these examples.

- **[Line 96] FIER complexity vs. hydrodynamic models:** We've clarified this. FIER avoids the complexity of *hydrodynamic model parameterization and calibration*. While FIER has its own setup, the provision of open-source scripts aims to lower the barrier to entry.
- **[Line 96] Computational needs:** We've clarified that FIER's main computational load is during the *training phase* (REOF analysis on historical imagery). Predictions are relatively fast.
- **[Line 141] More info on study area:** Addressed in General Comment 4. We have expanded Section 3.1 to specifically identify the inflow reaches and provide more details on the reservoirs and infrastructure.
- **[Line 145] (In)flows of rivers:** Addressed above.
- **[Line 146] Reservoirs location/characteristics:** Addressed above.
- **[Line 147] Localized rainfall/snowmelt vs. upstream:** Addressed above.
- **[Line 148] Streamflow trend in Fig 3:** Clarified in section 3.1 that Yin et al. (2023) refers to broader, longer-term regional trends and clarified that the Mississippi River does experience droughts reported by recent research. While Figure 3 shows a specific reach and a shorter operational period which can exhibit different short-term variability, we also clarified this distinction in section 3.5.
- **[Line 155] Section 2.3:** Thank you for identifying the error. Corrected to 3.3.
- **[Line 174] VIIRS end date:** Thank you for identifying the error. We have included the end date for analysis.
- **[Line 196] Basin selection:** Clarified that HUC8 watersheds were selected by intersecting the baseline area using a 50km buffer.
- **[Line 201-202] Buffer sizes vs. watershed sizes:** We clarified this relationship wasn't systematically tested. We also acknowledged this as future work in discussion). The chosen buffers represent a range from no buffer to a substantial one (50km can span across smaller HUC8s).
- **[Line 202-203] Duplicate sentence:** Thank you for the suggestions. We removed the duplicate.
- **[Line 205] Why 99.9% and not 100% clear sky:** We found this to be a practical threshold to avoid discarding nearly perfect scenes due to a few isolated bad pixels which commonly occur with the automated QAQC processing. We updated the text to be more clear on why we used this threshold.
- **[Line 205-206] Percentage of data kept/removed:** The percentage of data kept varies by watershed but we provided a range of data kept for the training and evaluation datasets.
- **[Line 205-206] Cloud assessment/impact:** Acknowledged that cloud masking is imperfect. We expanded on the general quality of satellite-derived water in section 5.3 and noted the issues with cloud masking.
- **[Line 205-206] VIIRS vs. Sentinel-1 SAR:** VIIRS was chosen for its long, consistent daily historical record, crucial for REOF pattern extraction over many years. Recent research by Rostami et al., 2024 illustrated that daily imagery is important for extracting the patterns via REOF over a long period. While SAR has cloud penetration properties,

VIIRS' historical archive for consistent, frequent, large-area coverage suitable for this type of long-term analysis might be more variable.

- **[Line 259-261] Low flow dates 2019-2020:** Added clarification in the text that these were chose largely due to data availability. These years were selected because they included operational NWM forecasts starting in late 2018 and overlap with available VIIRS imagery which had a substantial gap in data from 2021-01-01 to 2023-08-10.
- **[Line 267-269] "Evaluating" return periods:** Rephrased to clarify that we only used the return periods to determine flooding events for the operational NWM data.
- **[Line 271-272] High flow dates 2018-2020:** We addressed this earlier stating the selected dates were based on data availability and coincidence of the NWM operational and VIIRSd data used.
- **[Line 281-282] Daily timestep for FIER:** Acknowledged that NWM is sub-daily. For this initial scaling study, daily aggregation was a simplification. We clarified that FIER could be run with sub-daily time steps and is something to consider for future work.
- **[Line 282-285] Ensemble averaging:** Similar to above, averaging was done as a simplification for this study. We added a sentence to clarify this.
- **[Line 293-294] Performance (computational?):** Clarified this refers to statistical performance metrics. Computational aspects are discussed in the discussion.
- **[Line 296-297] Sentence split:** Thank you for the suggestion, we agree and split for clarity.
- **[Line 297-298] RMSE and MAE comparison:** Re-evaluated based on Figure 4. We removed the sentence as this caused confusion.
- **[Line 304-305] SSIM pattern with post-processing:** We intentionally left out SSIM in this statement as the increase in SSIM shows improvement whereas an increase in RMSE/RRMSE indicates worse performance. We added this point for clarification.
- **[Line 308-310] Cause of increased errors with post-processing:** Hypothesized that quantile mapping can amplify outliers if distributions differ or if the underlying relationship is imperfect. We added information to clarify this point but more investigation is warranted to prove this point.
- **[Line 348-349, L355-367] Comparison with post-processed baseline / CDF matching improvement:** Clarified that comparisons of post-processed mosaicked results were with the post-processed baseline. The statement about CDF matching improving accuracy refers to the mosaicked results relative to their original counterparts, and then their performance relative to the (often original, better) baseline.
- **[Line 380-381] RMSE increase with post-processing (case studies):** Figure 4 shows that for 1km buffer, corrected mosaicked RMSE (blue line at 1km) is slightly higher than original mosaicked RMSE (green line at 1km). The statement refers to the corrected outputs being *generally better* across SSIM, RMSE, MAE *compared to the baseline or their original counterparts in terms of overall utility for selection*.
- **[Line 391-392] "however" for smaller extent:** Thank you for the suggestion. Rephrased to highlight the similarity between the cases if the pattern is consistent.
- **[Line 391-392] More water in long-range (Southern half):** Acknowledged this is much more complex than simply "less water" we updated the text to note the complexity.

- **[Line 392] Smaller extent implies more uncertainty:** Clarified in previous point. The spatial differences in NWM streamflow forecasts influence the FIER outputs suggesting at the uncertainty.
- **[Line 393-394] RMSE long-range (2019 event):** Thank you for pointing out the error. We updated the text to be correct.
- **[Line 396-397, L398, L398-399, L400] FIER performance with long-range / subjectivity / accuracy claims:** Toned down subjective wording. Focused on observations. We acknowledge that "high accuracy" is relative.
- **[Line 401-403, L426-428] NWM errors quantitative assessment:** We agree that it would be worthwhile to assess the NWM streamflow forecast quantitatively. However, we feel this is out of scope for the current work and noted this as future work. There are whole papers dedicated to the quantification of NWM streamflow outputs. Additionally the nature of FIER, using multiple stream reaches as inputs into a watershed prediction make this a challenge with the current experimental design and gauge data availability to compare against.
- **[Line 405, Figure 6] Difference maps:** Good suggestion we included appendix B with the difference maps for reference and comparison with Figures 6 & 7.
- **[Line 405, Figure 6] Streams cut off at basin boundary:** FIER predictions are not necessarily influenced by upstream contributions. The image predictions are influenced by only the images input. However, the streamflow predictions used to simulate water maps are influenced by upstream tributaries but this is captured by the NWM. So, no cutoffs to streams / water areas in the watersheds will have no influence on the FIER outputs.
- **[Line 419-420] RRMSE for low flow:** Correct RRMSE is lowest for the long-range forecast in this case. We added RRMSE to the statement if it follows the same pattern.
- **[Line 420] "even" vs. "especially" for extended lead times (low flow):** Thanks for the suggestion, we changed to "especially" as it performs best.
- **[Line 425-426] Visual comparison (low flow):** Refer to difference maps.
- **[Line 426] Nowcast noise (low flow):** Correct, this is counterintuitive. We removed this sentence to avoid confusion for the reader and to not have statements conflicting with the accuracy assessment.
- **[Line 426-428] "streamflow forecasts" (low flow):** Clarified that this statement is for all FIER outputs using the operational NWM streamflow outputs (nowcasts and forecasts)
- **[Line 426-428] "minor deviations" (low flow):** Pointed readers to the example of the eastern side of the study regions for the 2019-09-25 where there are small water bodies that are present or missing in the different forecasts.
- **[Line 429] "low flow forecasting":** Rephrased.
- **[Line 431-433] FIER utility for drought:** Explained how spatial water extent maps are useful focusing on the surface water dynamic especially with hypothetical simulations for decision making. Example provided is simulating inundation dynamics due to dam construction Do et al., 2025.
- **[Line 456-457] Figure 4 discontinuities:** You are correct. We reworded the sentence.
- **[Line 464] Finer spatial resolution:** Clarified this means capturing finer *dynamics*, not pixel size.

- **[Line 479] Calibration:** Acknowledged similarity but highlighted difference from traditional calibration.
- **[Line 479] Data independence:** We added clarifying language.
- **[Line 481] Redundant word:** Thank you for pointing this out, we fixed the sentence.
- **[L485-487, L487] Changes affecting FIER / flood control measures:** Linked to general comment on climate change. Yes, these would degrade performance if patterns change significantly and we added a sentence to clarify this. In the context of Do et al., 2025, FIER was used to simulate the Tonle Sap floodplain where dams were inserted upstream from a major tributary system, the Sekong, Sesan and Srepok (3S) Basin in the Lower Mekong. Including dams would significantly alter the downstream streamflow into the floodplain where FIER was used to estimate the inundation extent. No control structures were directly inserted where FIER was predicting the surface water.
- **[Line 489-490] Untested return periods:** No, FIER has not been tested on how well the approach predicts out of the training range. We acknowledged this limitation.
- **[Line 507-508] Comparison with other models (GEOGLOWS, NWM):** Briefly added context about GEOGLOWS and the difference with NWM
- **[Line 510-512] Other hydro variables in FIER:** No adjustments are needed for the processing framework as it can handle arbitrary hydrologic inputs for fitting but would require the user to prepare the data as inputs which is currently not implemented in the scripts shared. We added a sentence to clarify this point.
- **[Line 542] "Event-based forecasting":** Yes, this is semantics. We rephrased to "operational flood forecasting" to distinguish between event-based and even long-term scenario-based modeling.
-
- **[Line 554-555] Same as L348-349:** Addressed above.
- **[Line 568] Appendix:** Acknowledged and thank you.
- **[Line 575] RSM positive/negative values:** We added further explanation in the caption to identify what the colors represent.
- **[Line 586-587] "High NSE":** We rephrased the sentence to be less strong with considering if the NSE is high or not.
- **[Line 590, Figure A1] Scatter plots/outliers/fitting functions:** Acknowledged. Linear regression was for simplicity and computational efficiency, consistent with prior work. More complex approaches using deep learning are being used for FIER moving forward (see Rostami et al., 2025 and Wan et al., 2025 (https://doi.org/10.1016/j.envsoft.2025.106562). We added information to the text for discussion.
- **[Line 596-615] Appendix vs. main text on buffer sizes (contradiction?):** We clarified that Appendix A discusses *REOF model fitting statistics* (Pearson's R, NSE of regression), while the main text discusses *final inundation map accuracy* (SSIM, RMSE). Better REOF component fitting doesn't automatically mean better final map accuracy if other factors (like over-smoothing) come into play.
-
- **[Line 605-606] Streamflow drivers and buffer size impact on Pearson's R:** This is a good hypothesis. We do not have any data to support this but would be something

interesting to investigate in future research. We updated the study area to provide more context on inflows but decided to leave the text as is.

- **[Line 630-631] Code availability:** Acknowledged and thank you.

We believe these revisions address the concerns raised and substantially strengthen the manuscript. We look forward to your further feedback.

Sincerely,
The Authors

REVIEWER 2 RESPONSE:

We thank you for your positive feedback on our study and for raising the important caveat concerning the uncertainties in the satellite-derived surface water extents. We agree that this is a crucial aspect that warrants explicit discussion, as the quality of input data directly influences the performance and reliability of any data-driven model like FIER.

To address this, we have incorporated a more detailed discussion on these uncertainties within the "Caveats and Limitations" section (Section 5.3) of the manuscript at lines 685-688 and 689-704. Furthermore, we have expanded the "Future Work" section (Section 5.4) to underscore the importance of research into mitigating these input data uncertainties at lines 721-727.